# Chemical composition and mixing state of BC-containing particles and the implications on light absorption enhancement

Jiaxing Sun[1,2], Yele Sun[1,2,3], Conghui Xie[1,2, a], Weiqi Xu[1], Chun Chen[1,2], Zhe Wang[1], Lei Li[4], Xubing Du[4,5], Fugui Huang[6], Yan Li[1,2], Zhijie Li[1,2], Xiaole Pan[1], Nan Ma[7], Wanyun Xu[8], Pingqing Fu[9], and Zifa Wang[1,2,3]

[1]State Key Laboratory of Atmospheric Boundary Layer Physics and Atmospheric Chemistry, Institute of Atmospheric Physics, Chinese Academy of Sciences, Beijing 100029, China

[2]College of Earth and Planetary Sciences, University of Chinese Academy of Sciences, Beijing 100049, China

[3]Center for Excellence in Regional Atmospheric Environment, Institute of Urban Environment, Chinese Academy of Sciences, Xiamen 361021, China

[4]Institute of Mass Spectrometry and Atmospheric Environment, Jinan University, Guangzhou, 510632, China

[5]Guangdong Provincial Engineering Research Center for On-Line Source Apportionment System of Air Pollution, Guangzhou, 510632, China

[6]Guangzhou Hexin Analytical Instrument Company Limited, Guangzhou, 510530, China

[7]Institute for Environmental and Climate Research, Jinan University, Guangzhou 511443, China

[8]State Key Laboratory of Severe Weather & Key Laboratory for Atmospheric Chemistry, Institute of Atmospheric Composition, Chinese Academy of Meteorological Sciences, Beijing, 100081, China

[9]Institute of Surface-Earth System Science, Tianjin University, Tianjin 300072, China

[a]now at: State Key Joint Laboratory of Environmental Simulation and Pollution Control, College of Environmental Sciences and Engineering, Peking University, Beijing, 100871, China

*Correspondence:* Yele Sun (sunyele@mail.iap.ac.cn)

**Abstract.** The radiative forcing of black carbon (BC) depends strongly on its mixing state in different chemical environments. Here we analyzed the chemical composition and mixing state of BC-containing particles by using a single particle aerosol mass spectrometer and investigated their impacts on light absorption enhancement ($E_{abs}$) at an urban (Beijing) and a rural site (Gucheng) in the North China Plain. While the BC was dominantly mixed with organic carbon (OC), nitrate and sulfate at both urban and rural sites, the rural site showed a much higher fraction of BC coated with OC and nitrate (36% vs. 15–20%). Moreover, the BC mixing state evolved significantly as a function of relative humidity (RH) with largely increased coatings of OC-nitrate and nitrate at high RH levels. By linking with organic aerosol (OA) composition, we found that the OC coated on BC comprised dominantly secondary OA in Beijing, while primary and secondary OA were similarly important in Gucheng. Furthermore, $E_{abs}$ was highly dependent on secondary inorganic aerosol coated on BC at both sites, while the coated primary OC also resulted in an $E_{abs}$ of ~1.2 for relatively fresh BC particles at the rural site. Positive matrix factorization analysis was performed to quantify the impact of different mixing state on $E_{abs}$. Our results showed the small $E_{abs}$ (1.06–1.11) for BC particles from fresh primary emissions, while the $E_{abs}$ increased significantly above 1.3 when BC was aged rapidly with increased coatings of OC-nitrate or nitrate, and it can reach above 1.4 as sulfate was involved in BC aging.

## 1 Introduction

Black carbon (BC), also referred to as soot or elemental carbon, contributes a substantial and positive impact on climate radiative forcing (Bond et al., 2013; IPCC, 2013). High loading of atmospheric BC could depress the development of the planetary boundary layer and aggravate the haze pollution episodes (Ding et al., 2016). However, accurate estimation of light absorption and radiative forcing of BC is still challenging due to its complex emission sources (e.g., fossil fuel and biomass burning) and microphysical properties (e.g., mixing state and coating composition) (Kahnert, 2010; Vignati et al., 2010; Liu et al., 2017; Cappa et al., 2019; Sun et al., 2022). Light absorption of BC is composed of pure BC absorption and the enhanced light absorption ($E_{abs}$) induced by the "lensing effect", which is due to the chemical materials coated on BC (Fuller et al., 1999; Bond and Bergstrom, 2006; Lack and Cappa, 2010). In order to determine the $E_{abs}$, the thermodenuder (TD)-method (Liu et al., 2015; Zhang et al., 2016), mass absorption efficiency (MAE)-method (Wang et al., 2014; Zhang et al., 2018) and Mie

theoretical calculations (Liu et al., 2017) are widely used in previous studies. However, the $E_{abs}$ in different chemical environments is considerably different, such as the negligible $E_{abs}$ (1.06) in California versus a significant light absorption enhancement (~1.8) in Kanpur (Cappa et al., 2012; Thamban et al., 2017). One explanation is due to the variety of the mixing state of BC from different sources, urban and rural background sites, and aging processes (Liu et al., 2015; Liu et al., 2017; Liu et al., 2020).

Extensive studies have been conducted to characterize the mixing state of BC (Li et al., 2016b). Buseck et al. (2014) used transmission electron microscopy (TEM) to determine the size, morphology, mixing state and chemical compositions of soot particles, and Li et al. (2016a) used the same technique for the characterization of the size and aging dependent mixing state of individual particles in both clean and polluted environments in China. More recently, the development of the soot particle aerosol mass spectrometer (SP-AMS) is able to measure refractory BC (rBC) and coated aerosol species in real-time (Lee et al., 2015). For example, Xie et al. (2019a) found that secondary organic and inorganic components contributed mostly to the coating materials of rBC in summer, while Liu et al. (2019a) reported that rBC was more mixed with primary organic components (e.g., coal combustion and biomass burning) than secondary species in winter. Comparatively, a more comprehensive mixing state of BC particles and the chemical compositions of coatings could be acquired by aerosol time-of-flight mass spectrometry (ATOFMS) or single particle aerosol mass spectrometry (SPAMS) (Pratt and Prather, 2012). Recent studies in winter in the North China Plain (NCP) showed that most BC particles were mixed with sulfate and organic carbon (OC) due to the intensive coal combustion and biomass burning emissions in the heating period (Chen et al., 2020b; Wang et al., 2020), while BC particles were more mixed with nitrate in summer in Beijing (Xie et al., 2020). Although the mixing state, coating compositions and $E_{abs}$ of BC have been widely discussed in previous studies, our understanding of BC properties in winter, particularly the differences between urban and rural areas in the NCP, are limited.

In this study, we deployed a newly developed high resolution single particle aerosol mass spectrometer (Zhu et al., 2020), and a photoacoustic extinctiometer (PAX) coupled with a thermodenuder at an urban and a rural site in the NCP to investigate the mixing state of BC-containing particles and its relationship with light absorption enhancement. Meanwhile, the aerosol bulk composition was measured simultaneously by a high-resolution aerosol mass spectrometer (HR-AMS). The chemical composition and mixing state of BC-containing particles are characterized, and the evolution of the BC mixing state and the

driving factors are investigated. Finally, the impacts of the changes in the mixing state of BC on light absorption enhancement and radiative forcing are elucidated.

## 2 Methods

### 2.1 Sampling sites and measurements

The measurements were conducted at an urban site from 18 October to 1 December, 2019 and a rural site from 9 December 2019 to 13 January 2020. The urban site is located at the Institute of Atmospheric Physics (IAP), Chinese Academy of Sciences (39°58′28″N, 116°22′16″E) in Beijing (BJ). The rural site of Gucheng (GC) in Hebei province is located ~120 km to the southwest of Beijing. Ambient aerosols with a flow rate of 3 L min$^{-1}$ were drawn into a PM$_{2.5}$ cyclone (Model: URG-2000-30ED) and a nafion dryer, then aerosol particles (~2 L min$^{-1}$) were sampled by switching between a thermodenuder (TD) and a bypass line (BP) every 15 min into a HR-AMS to measure non-refractory submicron aerosol (NR-PM$_1$) species and a PAX (Droplet Measurement Technologies) for absorption [$b_{abs}(\lambda)$] and scattering [$b_{sca}(\lambda)$] coefficients at 870 nm. Note that equivalent BC (eBC) is converted from $b_{abs}(\lambda)$ into mass concentration with a reference mass absorption efficient (MAE$_{ref}$, 4.44 m$^2$ g$^{-1}$ in this study) (Bond and Bergstrom, 2006). A complete TD cycle took 150 minutes, during which the temperature increased gradually from 25°C to 250°C followed by a 15 min cooling down to 25°C. To minimize the uncertainties due to the changes during the measurements of TD and bypass, the $b_{abs, total}$ was obtained from the linear interpolation of measured ambient absorptions adjacent to the TD time, and the $E_{abs}$ was then determined as the ratio of $b_{abs, total}$ to $b_{abs, BCpure}$ that was defined as the thermodenuded particle absorption in the TD line at $T > 200$°C. Note that we did not do TD loss correction in this study. The reason is that the underestimated $E_{abs}$ of ~14% due to the TD loss (Fig. S1) can be relatively offset by the overestimation of ~16% due to the coating materials that did not evaporate at 210°C (Ma et al., 2020; Xu et al., 2021a). More detailed descriptions of the measurements and methods could be found in our previous study (Sun et al., 2021).

A high-resolution single particle aerosol mass spectrometer (HR-SPAMS, Hexin Instrument Co., Ltd.) was deployed independently in the same room for detecting the mixing state and chemical composition of single particles in the ambient atmosphere. The aerodynamic diameters of single particles are determined from the velocity detected by two continuous laser

beams (diode Nd: YAG, 532 nm). After passing through the sizing region, particles are desorbed and ionized by a pulsed Nd: YAG laser (266 nm), and the positive and negative fragments are detected by a Z-shaped bipolar time-of-flight mass spectrometer to obtain the chemical compositions. A more detailed description of the SPAMS can be found elsewhere (Li et al., 2011). Compared with the traditional SPAMS, the new HR-SPAMS uses a new aerosol concentration sampling device and improves the transmission efficiency of coarse particles. Meanwhile, a delayed extraction technology is introduced to improve the mass resolution and increase the hit rate by a factor of 2–4 for ambient particles (Chen et al., 2020c). Also, the ion signals with high and low intensity are separated by multichannel acquisition technology and detected simultaneously, which makes the system dynamic range more than 40 times of the traditional data acquisition system and improves the detection of ions with low signals greatly (Shen et al., 2018; Zhong et al., 2021).

## 2.2 Data analysis

In total, 3619038 and 4655426 particles were analyzed in BJ and GC, respectively. The size and chemical composition of each single particle is informed by the Computational Continuation Core (COCO) toolkit in the MATLAB software. Based on the marker of $C_n^{\pm}$ (n = 1, 2, 3…) clusters, 2269 659 and 3399565 BC-containing particles were identified in BJ and GC, respectively. Four typical sources of BC-containing particles are further identified according to the characteristic ion markers: (1) particles containing abundant signals of $39[K]^+$, $45[CHO_2]^-$ and $59[C_2H_3O_2]^-$ or $73[C_3H_5O_2]^-$ with peak areas more than 0.5% are classified as $BB_{pure}$ type from biomass burning (Silva et al., 1999; Healy et al., 2010); (2) particles containing abundant signals of $7[Li]^+$, $23[Na]^+$, $27[Al]^+$, $43[AlO]^-$, $80[SO_3]^-$, $97[HSO_4]^-$ and polycyclic aromatic hydrocarbons are classified as $CC_{pure}$ type from coal combustion (Zhang et al., 2009; Healy et al., 2010); (3) particles containing abundant signals of $40[Ca]^+$, $51[V]^+$, $55[Mn]^+$, $67[VO]^+$, $46[NO_2]^-$, $62[NO_3]^-$ and $79[PO_3]^-$ are classified as $TR_{pure}$ type from traffic emissions (Yang et al., 2017); (4) particles internally mixed with more than one source (three sources above) are unified together and named as $Mix_{Source}$ type. The BC-containing particles above are collectively referred to $BC_{fresh}$. The remain BC-containing particles are named $BC_{aged}$ and classified by using the ART-2a algorithm with a vigilance factor of 0.75, a learning rate of 0.05 and 20 iterations (Song et al., 1999). Seven particle types are grouped and named based on two principles: (1) the particles are named BCOC when the signals of $37[C_3H]^+$, $43[C_2H_3O]^+$, $51[C_4H_3]^+$ and $63[C_5H_3]^+$ are comparable with $C_n^+$ in the positive

ion area (Healy et al., 2012; Xie et al., 2020), otherwise they are named BC; (2) On the basis of (1), the particles are named BCOC$_N$ or BCOC$_S$ when they are only mixed significantly with nitrate (46[NO$_2$]$^-$ and 62[NO$_3$]$^-$) or sulfate (97[HSO$_4$]$^-$). Otherwise, they are named BCOC$_{NS}$ when they present comparable peak areas of nitrate and sulfate. The more detailed names of BC-containing particle types are given in Table S1. According to previous studies, the coating materials on BC-containing particles measured by SPAMS referred to chemical components that are partially or fully coated on BC (Bond and Bergstrom, 2006; Healy et al., 2012; Pratt and Prather, 2012; Bi et al., 2015; Chen et al., 2016; Xie et al., 2020).

The sources of bulk organic aerosol (OA) from HR-AMS measurements were analyzed by positive matrix factorization (PMF) and five OA factors were identified at both urban and rural sites, including biomass burning OA (BBOA), fossil fuel-related OA (FFOA), cooking OA (COA), less oxidized oxygenated OA (LO-OOA) and more oxidized OOA (MO-OOA) in Beijing and BBOA, coal combustion OA (CCOA), hydrocarbon-like OA (HOA), OOA and aqueous-related OOA (aq-OOA) in GC. The detailed PMF analysis of OA in Beijing and Gucheng are presented by Xu et al. (2021b) and Chen et al. (2021). The PMF analyses was also performed to identify the effect of different mixing state on $E_{abs}$, by inputting $b_{abs, total}$, $b_{abs, BCpure}$ and 11 major types of BC-containing particles derived from HR-SPAMS. The detailed pre-treatment of the error matrix (in the Supplementary), and the selection of factor solutions could be found in previous studies (Petit et al., 2014; Xie et al., 2019b). Then, the $E_{abs}$ of each factor can be calculated as the ratio of $b_{abs, total\ fi}$ and $b_{abs, BCpure\ fi}$ in factor $i$.

## 3 Results and discussion

### 3.1 BC-containing particles at urban and rural sites

The BC-containing particles accounted for 62% of the total particles in BJ, lower than in GC (73%) yet higher than in winter Beijing 2018 (55%) (Xie et al., 2020) likely due to the higher mass fraction of eBC in this study (9.3% vs. 6.1%). Similarly, a previous winter study in Beijing also found that 60–78% of the aerosol particles contained BC (Chen et al., 2020a). According to Figs. 1 and S2, we found that the mass spectra of the BC-containing particles at the two sites are somewhat similar which are both characterized by C$_n$$^\pm$ (n = 1–7), 27[C$_2$H$_3$]$^+$, 37[C$_3$H]$^+$, 43[C$_2$H$_3$O]$^+$, 51[C$_4$H$_3$]$^+$ 63[C$_5$H$_3$]$^+$, 46[NO$_2$]$^-$, 62[NO$_3$]$^-$ and 97[HSO$_4$]$^-$ peaks, indicating that the BC-containing particles are consistently mixed with OC, nitrate (NO$_3$) and sulfate (SO$_4$)

at the rural and urban sites. Comparatively, more than 80% of the BC-containing particles were internally mixed with $SO_4$ in BJ while those in GC accounted for less than 60%. Considering the higher relative area of the secondary organic fragment of $43[C_2H_3O]^+$ (Healy et al., 2010; Chen et al., 2019) in BJ (Fig. 1), we concluded that the BC particles were likely more aged at

145   the urban site. Another support is much lower primary emissions of biomass burning and coal combustion in BJ than GC (Sun et al., 2020). In addition, approximately 40% of the BC-containing particles was mixed with amines (e.g., $59[(CH_3)_3N]^+$ and $74[(C_2H_5)_2NH_2]^+$ ) in GC, which was twice that in BJ. One explanation is the more nitrogen-containing compounds formed from either aqueous-phase processing due to higher RH (69% vs. 45%) or biomass burning emissions at the rural site (Zhang et al., 2012; Chen et al., 2019).

150   The study in Beijing was divided into two periods, i.e., non-heating period (BJ-NHP) from 28 October to 10 November and heating period (BJ-HP) from 10 November to 1 December, while the observation in GC was performed during heating period. As illustrated in Fig. 2a, the BC-containing particles are dominantly contributed by $BC_N$ and $BCOC_N$ (~20%) during BJ-NHP, indicating that BC was mainly internally mixed with nitrate at the urban site. Compared with BJ-NHP, the fractions of $BCOC_S$ and $Mix_{Source}$ increased significantly during BJ-HP, especially during relatively clean periods, suggesting a considerable change

155   in BC mixing state from non-heating to heating period due to the enhanced primary emissions, e.g., coal combustion. Previous studies showed that BC was mainly mixed with sulfate in winter in Beijing (Chen et al., 2020b; Xie et al., 2020), while this study showed a dominant mixing of BC with nitrate. This was likely due to the fact that coal fuels in Beijing were replaced by clean energies, e.g., natural gas and electricity after 2017 (Zhang et al., 2019). Indeed, the changes in nitrate concentrations were relatively small in winter in Beijing since clean air action although the sulfate concentrations showed large decreases

160   (Zhou et al., 2019; Lei et al., 2020). Comparatively, $BCOC_N$ was the major BC-containing particle type accounting for 36% in GC, which was twice that in BJ-HP indicating that BC particles were dominantly mixed with OC and nitrate in an environment with high RH and intensive primary emissions, e.g., coal combustion emissions. In addition, $BB_{pure}$ and $TR_{pure}$ showed pronounced diurnal cycles in GC (Fig. S3) compared with the relatively flat diurnal variations of $BB_{pure}$ in BJ, suggesting intensive biomass burning and diesel vehicle emissions at the rural site especially at nighttime.

## 3.2 Chemical composition and mixing state in different environment

Figure 3 shows the variations of number fractions of different BC-containing types under different RH levels in BJ and GC. Almost all $BC_{aged}$ types showed strong dependence on RH while the number fractions of $BC_{fresh}$ types decreased with increasing RH at both sites indicating that a high RH environment was more favorable for BC aging (Zhang et al., 2021). Similar to $BC_{fresh}$, $BCOC_S$ was the only type of aged BC showing a decreased fraction as a function of RH in BJ and GC. In fact, the high correlations between $BCOC_S$ and $CC_{pure}$ ($R^2 = 0.89$ and 0.98, in BJ and GC, respectively) highlight that BC emitted from coal combustion could be directly mixed with OC and $SO_4$ at low RH level, and evolved towards the mixing with OC and $NO_3$ under high RH conditions. Moreover, the number fraction of $BC_N$ increased gradually with the increase of RH and dominated BC particles (25–30%) at RH = 70–100% in BJ. Considering the similar increases of PM (NR-PM$_1$ + eBC) as a function of RH (Figs. 3a and S4), the RH dependence of $BC_N$ suggested that the newly formed nitrate that coated on fresh BC played an important role in the formation of severe pollution in the urban region. Comparatively, the number fraction of $BCOC_N$ increased the most by 43%, accounting for more than half of the BC-containing particles at high RH and PM levels in GC. This result indicates that the type of $BCOC_N$ was more important to aggravate air pollution in the rural area, supported by the relatively high correlation between the number fraction of $BCOC_N$ and PM (Fig. S4). In addition, $BCOC_{NS}$ was likely affected by the photochemical production in GC, which is supported by the significantly increased number counts and fraction of $BCOC_{NS}$ during daytime (Fig. S3). We found that the fraction of $BCOC_{NS}$ decreased obviously as a function of RH in GC indicating the impact of the transition from photochemical production to aqueous-phase reactions on the mixing state of BC. Considering the increased $SO_4$ mass fraction yet the relatively stable $E_{abs}$ at RH > 70% (Sun et al., 2021), we inferred that aqueous-phase formation of sulfate at high RH level appeared not to affect the BC mixing state substantially, consistent with a previous study (Zhang et al., 2021). However, $BCOC_{NS}$ in BJ showed relatively stable fractions across different RH levels suggesting the different sources from the rural site.

Figs. 3c and 3d show the evolution of the mixing state of BC-containing particles during two different haze events in BJ. During the initial stage of haze case1 (P0, Fig. 3c), the contribution of $BCOC_S$ started to decrease while the number fraction of $BCOC_N$ increased significantly. As a consequence, $E_{abs}$ increased rapidly from ~1.1 to 1.3 in half a day. Then, the number

fraction of $BC_N$ increased while that of $BC_{fresh}$ decreased during the P1 period. These results indicated that fresh BC was gradually aged by mixing with nitrate during the evolution of the haze episode. $BC_N$ increased continually during P2 with $E_{abs}$ up to 1.4, and finally the mixing state of BC was stabilized as indicated by the relatively stable number fractions of most BC particle types and small changes in $E_{abs.}$ Similar to haze case1, BC was mixed with nitrate and OC causing a high $E_{abs}$ (up to 1.5) during P3 and P4 (Fig. 3d). As shown in Figs. 3d and S5, the sources of fine particles were dominated by fossil fuel OA and presented strong diurnal variations consistent with changes in wind direction due to the influences of mountain valley winds (Sun et al., 2016). As a result, $E_{abs}$ also presented a relatively consistent variation with FFOA and showed higher values at nighttime, indicating that the different chemical composition and mixing state associated with the changes in air masses due to the mountain valley winds had affected the light absorption enhancement of BC (Ding et al., 2021). The number fraction of $BC_N$ increased significantly during the severest polluted period (P5) associated with simultaneous increases in BBOA, indicating the mixing of BC from biomass burning with nitrate under high RH and PM levels, and hence $E_{abs}$ was comparably high (1.35). After the P5 period (from 21:00 on 9 November to 0:00 on 10 November), the BBOA decreased slightly and FFOA increased significantly. The proportion of $BCOC_N$ in BC was correspondingly higher. A similar variation could also be found during 14:00–19:00 on 8 November. These results indicated that BC emitted from fossil fuel emission was likely mixed with OC and nitrate at high PM level. Different from Beijing, the evolution of BC mixing state was similar during most of the haze events in GC (Figs. 2c and S3), which was generally characterized by a more significant increase in $BCOC_N$ than $BC_N$. Moreover, $BCOC_N$ particles increased more significantly during nighttime while $BCOC_{NS}$ was more significant during daytime. Such differences were mainly due to the enhanced coal combustion pollutants at nighttime (Fig. S3) which were mixed with photochemical products during daytime.

Overall, our results suggest that the fresh BC particles from biomass burning emissions are more directly mixed with nitrate under high RH conditions, and then mixed with more sulfate during further aging. Comparatively, the fresh BC particles from coal combustion are often mixed with OC and sulfate first, and then mix further with OC and nitrate at high RH level. In addition, our results also demonstrate that the $E_{abs}$ in GC was largely due to coal combustion emissions internally mixed with OC, consistent with our previous study showing a large impact of coal combustion emissions on $E_{abs}$ (Sun et al., 2021).

### 3.3 Effects of chemical composition on $E_{abs}$

As shown in the average positive mass spectra of the total BC-containing particles (Fig. S2), the peak areas of $C_n^+$, OC and metal contributed more than 95% to the total peak area, while the peak areas of $NO_3$ (46[$NO_2$]$^-$ and 62[$NO_3$]$^-$) and $SO_4$ (97[$HSO_4$]$^-$) accounted for more than 80% in the negative mass spectra. To better characterize the relationship between the chemical species and $E_{abs}$, we summed the $C_n^\pm$ (n = 1–5, accounting for more than 99% in $C_n$) peak areas to represent BC and the total of $NO_3$ and $SO_4$ peak areas to represent the secondary inorganic components coated on BC. In addition, the sum of the positive peak areas except $C_n^+$ was defined as OC + Metal to represent the OC and metal components coated on BC. These peak areas covered almost all of the chemical components coated on BC in the total BC-containing particles.

Figures 4a and 4b show the relationship between the peak area ratios (measured by HR-SPAMS) and the mass concentration ratios (measured by HR-AMS) in BJ and GC, respectively. With the increase of ($NO_3$ + $SO_4$)$_{AMS}$/eBC mass concentration ratio, the ($NO_3$ + $SO_4$)/$C_n$ peak area ratio increased first and then gradually became stable at both sites. These results indicated that BC was rapidly aged and internally mixed with secondary inorganic components during the early stage of the haze episode, and appeared to be fully aged when the mixing efficiency of $NO_3$ and $SO_4$ with BC reached a maximum, i.e., ($NO_3$ + $SO_4$)$_{AMS}$/eBC = ~6. Different from secondary inorganic species, the peak area ratio of (OC + Metal)/$C_n$ showed a high dependence on the mass concentration ratio of POA (e.g., the sum of BBOA and FFOA in BJ and the sum of BBOA, CCOA and HOA in GC) to eBC at both sites. These results indicated that the primary OA (POA) co-emitted with BC was more easily internally mixed with BC than secondary inorganic components (e.g., $SO_4$ and $NO_3$) although the mass concentration was much lower than that of secondary inorganic aerosol at both the urban and rural sites. Moreover, the mass concentration ratio of SOA/eBC also presented a high correlation with (OC + Metal)/$C_n$ in GC ($R^2$ = 0.81) and BJ ($R^2$ = 0.95, Fig. S7). We then used multiple linear regression analysis to quantify the impacts of POA and SOA on OC coated on BC. Our results showed that the average contribution of SOA to the coated OC was nearly twice that of POA (65% vs. 35%) in BJ, while POA and SOA contributed similarly in GC.

Figures 4c and 4d show the relationship between $E_{abs}$ and the ratio of mixing materials to $C_n$ in BJ and GC, respectively. The light absorption enhancement showed a strong dependence on ($NO_3$ + $SO_4$)/$C_n$ at both sites. Previous studies found that SOA

played an important role in the BC absorption enhancement (Liu et al., 2019b), whereas in our study, the changes in $E_{abs}$ seemed to be independent on $(OC + metal)/C_n$, likely because $(OC + metal)/C_n$ was influenced by both primary and secondary factors. As shown in Fig. 4c, the ratio of $(OC + Metal)/C_n$ still presented high values at $E_{abs} = \sim1$, while the secondary species coated on BC were negligible. These results suggested OC and metals likely either filled internal void spaces of fresh BC or mainly partly mixed with BC which did not induce light absorption enhancement at the urban site. With the progress of aging, $E_{abs}$ increased significantly mainly due to the increased secondary coating materials. Similar to BJ, $E_{abs}$ also increased significantly as a function of $(NO_3 + SO_4)/C_n$ in GC. The difference is the high background $E_{abs}$ of ~1.20 in GC when the $(NO_3 + SO_4)/C_n$ ratio was close to 0. A previous study showed that $E_{abs}$ is > 1 when the non-BC material is sufficient to encapsulate the BC (Liu et al., 2017). Considering the peak area ratio of $(OC + Metal)/C_n$ was ~2.5, we inferred that OC and metals were not only as filler materials but also likely coated on fresh BC and induced light absorption enhancement at the rural site. After the further aging process in the atmosphere, $E_{abs}$ was mainly due to the increased secondary inorganic components coated on BC. Moreover, we predicted the $E_{abs}$ using the statistical equations in Fig. 4 with aerosol species measured by AMS. We found that the estimated $E_{abs}$ showed overall agreements with the measured values in both BJ and GC. Although the correlation was not significant (Fig. S8), the average measured and estimated $E_{abs}$ values were similar which are 1.21 (±0.12) and 1.22 (±0.18), respectively in BJ, and 1.31 (±0.15) and 1.25 (±0.07), respectively in GC. Also, the uncertainty estimated from the difference of measured and predicted was overall below 10% in both BJ and GC indicating that the approach is reasonably well to estimate $E_{abs}$. We also estimated the $E_{abs}$ in summer 2017 using the same method and compared with the measurements by a cavity attenuated phase shift single scattering albedo monitor coupled with a thermodenuder (Fig. S9). Our results showed that the average $E_{abs}$ in summer was 1.24, which was close to the average of about 1.2 reported by Liu et al. (2019a), yet lower than that in Xie et al. (2019a).

### 3.4 Effects of mixing state on $E_{abs}$

The PMF analysis is used to characterize the effects of different mixing states on $E_{abs}$. Five and four factors were identified in BJ and GC, respectively, to elaborate the influence of different mixing states on $E_{abs}$ (Fig. 5). Note that $E_{abs}$ was not estimated when the factor contributed negligibly to the total BC, such as Factor5 in BJ (Fig. 5e). As illustrated in Fig. 5b, Factor2 is the

major type of aged BC in the urban region, accounting for more than 60% of the total BC. This factor was dominated by $BC_N$, $BCOC_N$, $BCOC_{NS}$ and $BC_{NS}$, and presented a high $E_{abs}$ of 1.38. Comparatively, FactorB (Fig. 5g) is the major type of aged BC in the rural area which is dominated by $BCOC_N$. The $E_{abs}$ was ~1.35 for this factor, which was comparable to that in BJ. As this factor evolved towards FactorA (Fig. 5f) after further aging and became internally mixed with a large amount of sulfate,

$E_{abs}$ was increased up to 1.41. In BJ, the relatively fresh traffic emissions are dominant in Factor4 (Fig. 5d) which showed a negligible impact on light absorption enhancement (~1.06), consistent with the results in previous studies (Liu et al., 2017; Sun et al., 2021). Compared with traffic emissions, the relatively fresh biomass burning and coal combustion emissions (Fig. 5c) comprising mainly $Mix_{Source}$ and $BCOC_S$ showed a moderate $E_{abs}$ (~1.11) in BJ. Although the mixed fresh primary emissions (Fig. 5i) in GC presented a relatively low $E_{abs}$ (1.06), we found that the FactorC from coal combustion emissions

and mixed with much OC and nitrate showed a much higher $E_{abs}$ (~1.31) than that in BJ (Fig. 5c). After aging, more sulfate could be internally mixed with BC and enhanced the $E_{abs}$, such as Factor1 (Fig. 5a) with $E_{abs}$ up to 1.42. Overall, $E_{abs}$ shows a similar dependence on the evolution of the mixing state of BC-containing particles at urban and rural sites, i.e., fresh BC particles from primary emissions (e.g., biomass burning, coal combustion and traffic) showed the small $E_{abs}$ (1.06–1.11). At relatively high RH level, BC could be directly mixed with nitrate or OC-nitrate ($BC_N$ and $BCOC_N$), accounting for more than

60% of BC, and lead to an increase in $E_{abs}$ above 1.30; and then, the BC-containing particles were further mixed with sulfate ($BC_{NS}$ and $BCOC_{NS}$) after continuous aging in atmosphere, and resulted in the highest $E_{abs}$ above 1.40.

Based on $E_{abs}$ for each factor and its contribution to $b_{abs, BCpure}$, we estimated the direct radiative forcing ($\Delta F_R$) caused by pure BC at the top of the atmosphere (TOA) and the absorption $\Delta F_R$ enhanced by the mixed state of BC-containing particles (Chylek and Wong, 1995; Chen and Bond, 2010). The detailed descriptions for the estimation are presented in the Supplementary. It

should be noted that BrC can also absorb light at 870 nm, leading to an overestimation of BC absorption. Considering that the contribution of BrC to the total absorption at 870 nm is typically small (< 1%) (Clarke et al., 2004; Fialho et al., 2005; Yang et al., 2009), the impact of BrC on the estimation of the radiative forcing of BC is expected to be small as well. As shown in Fig. 5, the $\Delta F_R$ caused by pure BC particles is about +0.43 W m$^{-2}$ and +0.60 W m$^{-2}$ in Beijing and Gucheng, respectively. Considering the mixing state of BC, $\Delta F_R$ could increase to 0.59 W m$^{-2}$ and 0.82 W m$^{-2}$ in Beijing and Gucheng, respectively.

Our results demonstrated that the mixing state of BC-containing particles can have a large impact on the radiative forcing estimation by up to 27%.

## 4 Conclusions

HR-SPAMS, TD-PAX and HR-AMS were deployed at the urban and rural sites in the North China Plain in winter 2019 to characterize the chemical composition and mixing state of BC-containing particles and their impacts on light absorption

enhancement. Our results showed that BC-containing particles were primarily mixed with OC, $NO_3$ and $SO_4$ at both sites while the rural site showed a much higher fraction of OC and nitrate coated BC (36% vs. 15–20%). The increased BC particles internally mixed with a large amount of $NO_3$ was mainly due to the effect of the clean air action that reduced much more sulfate than nitrate in $PM_{2.5}$. The average contribution of SOA to the OC coated on BC was about twice that of POA in BJ, while both of them contributed similarly to the OC coating in GC. In addition, OC and metals were likely as filler materials internally

mixed with fresh BC and did not induce much light absorption enhancement at the urban site, while they were coated on fresh BC and induced light absorption enhancement (~1.2) at the rural site.

By analysing the variations of the BC mixing state in different environments, we found that BC particles were primarily mixed with $NO_3$ with the increase of RH at both the urban and rural sites. In particular, the BC emitted from biomass burning was first mixed with $NO_3$ at relatively high RH level and then mixed with both nitrate and sulfate after further aging. Comparatively,

the BC emitted from coal combustion was more internally mixed with OC and $NO_3$, and then mixed with sulfate ($BCOC_{NS}$) with similar processes. Thus, high coal combustion emissions can result in the increase of the $BCOC_N$ fraction with the increase of RH at the rural site, while $BC_N$ was generally the dominant BC-containing particle type in Beijing. Although BC particles presented a different mixing state in different environments, $E_{abs}$ showed a similar evolutionary dependence on the changes in mixing state, i.e., from the small $E_{abs}$ (1.06–1.11) with fresh BC emissions to above 1.30 after aging and internal mixing with

nitrate and OC-nitrate ($BC_N$ and $BCOC_N$), and to above 1.40 after further aging with the sulfate involved.

*Data availability.* The data in this study are available from the authors upon request (sunyele@mail.iap.ac.cn).

*Author contributions.* YS and JS designed the research. JS, CX, WX, CC and ZL conducted the measurements. JS, WX, CC, ZW and YL analyzed the data. CX, LL, XD, FH, XP, NM, WX, PF and ZiW reviewed and commented on the paper. JS and YS wrote the paper.

*Competing interests.* The authors declare that they have no conflict of interest.

*Acknowledgements.* This work was supported by the National Natural Science Foundation of China (9204430003, 42061134008).

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

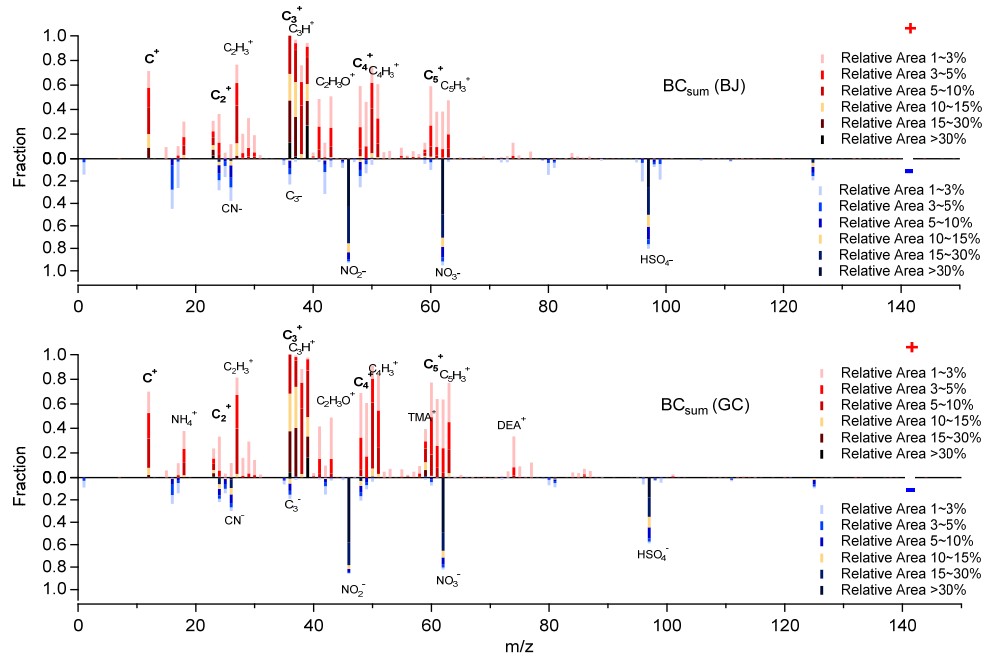

**Figure 1. Digital positive and negative mass spectra of BC-containing particles in Beijing and Gucheng. The ion height indicates its number fraction in the BC-containing dataset (i.e., the number ratio of BC-containing particles with the corresponding ion detected in the mass spectra to the total BC-containing particles). The color bars represent each relative peak area corresponding to a specific fraction in the individual particles.**

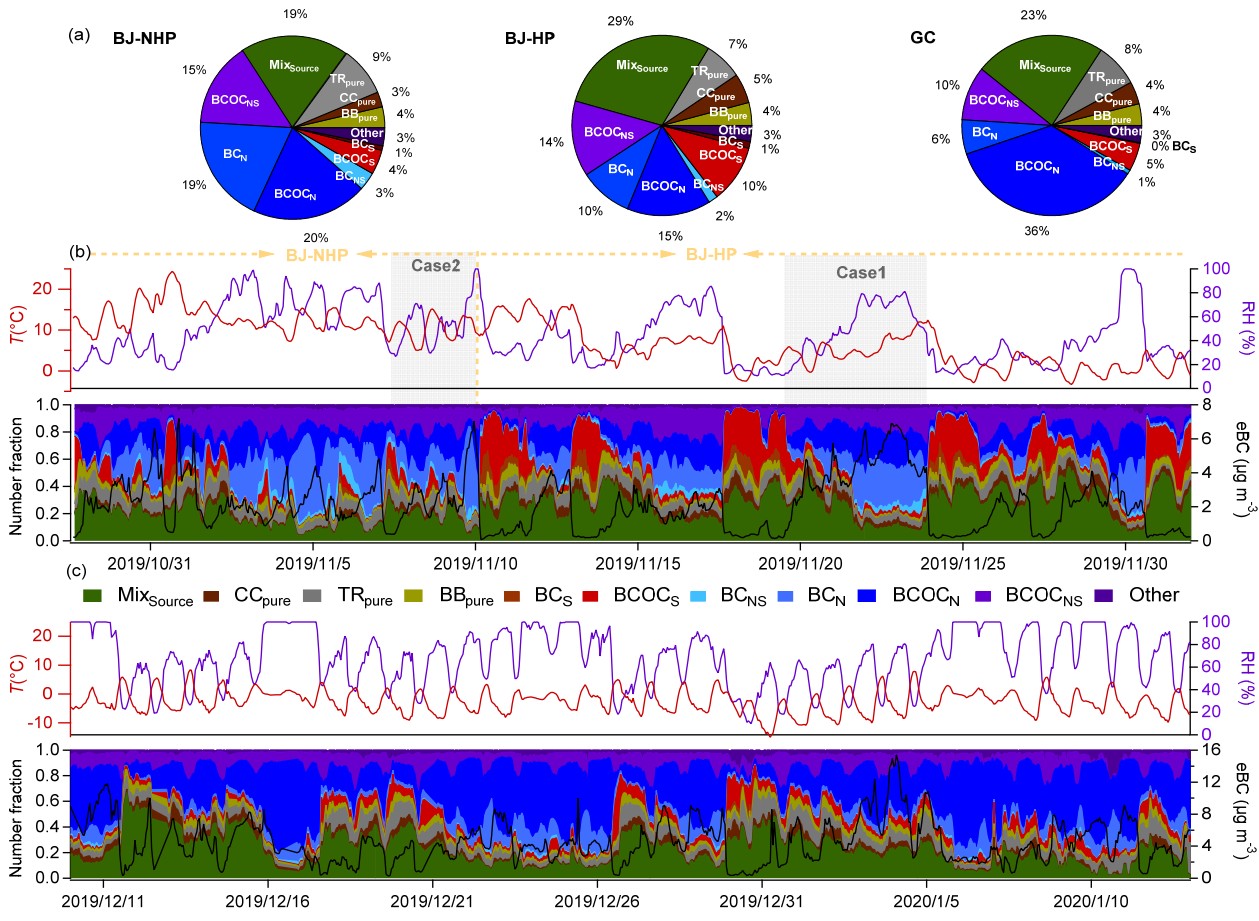

**Figure 2. (a) Relative number abundance distribution of major BC-containing particle types during different periods. Temporal variation of ambient temperature (*T*), relative humidity (RH) and number fractions of BC-containing particle types in (b) Beijing and (c) Gucheng.**

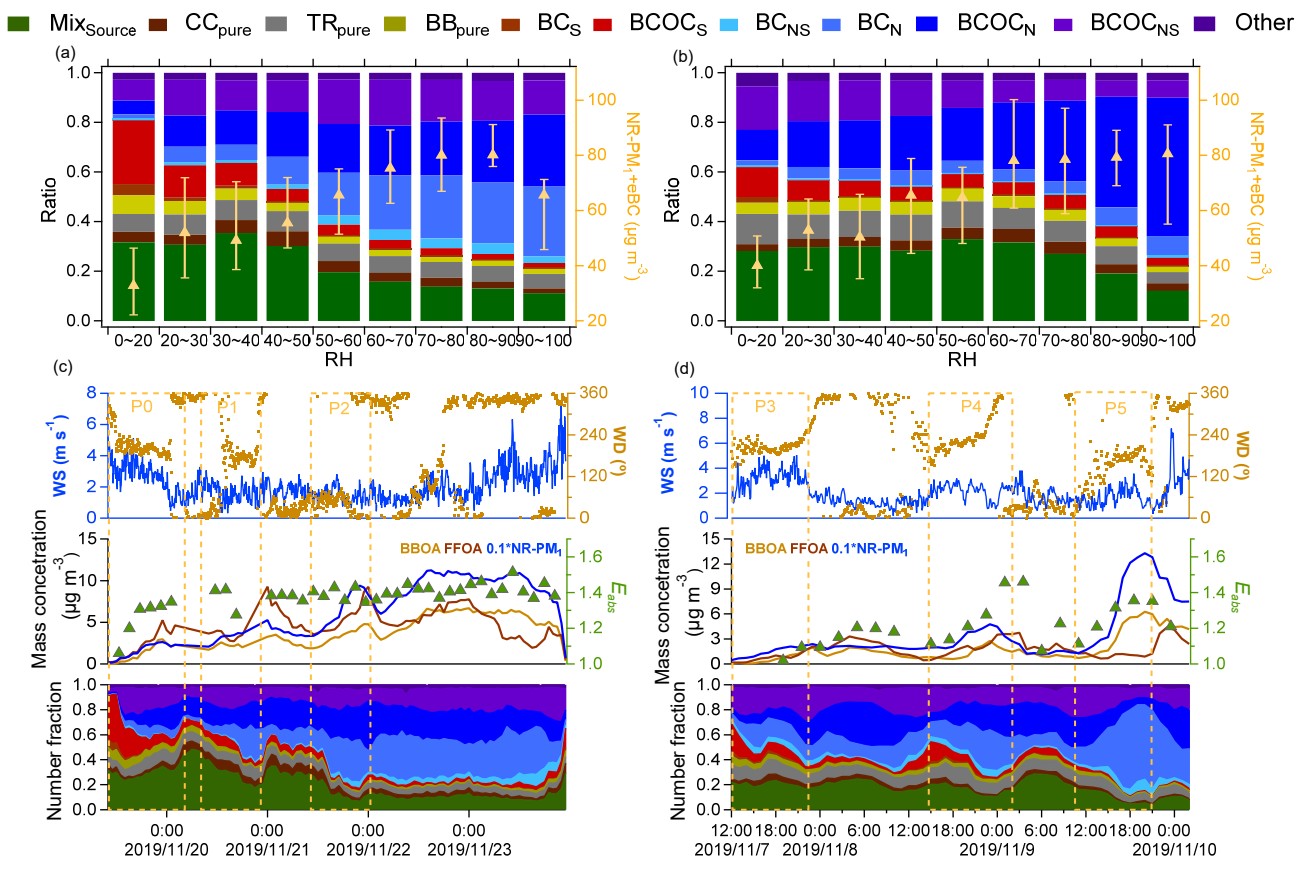

**Figure 3.** Variations of number fractions of BC-containing particle types with RH in (a) Beijing and (b) Gucheng. The bottom and top error bars represent 25th and 75th percentiles. Temporal variations of wind speed (WS), wind direction (WD), $E_{abs}$, number fractions of BC-containing particle types and mass concentration of species during polluted (c) Case1 and (d) Case2 in Beijing.

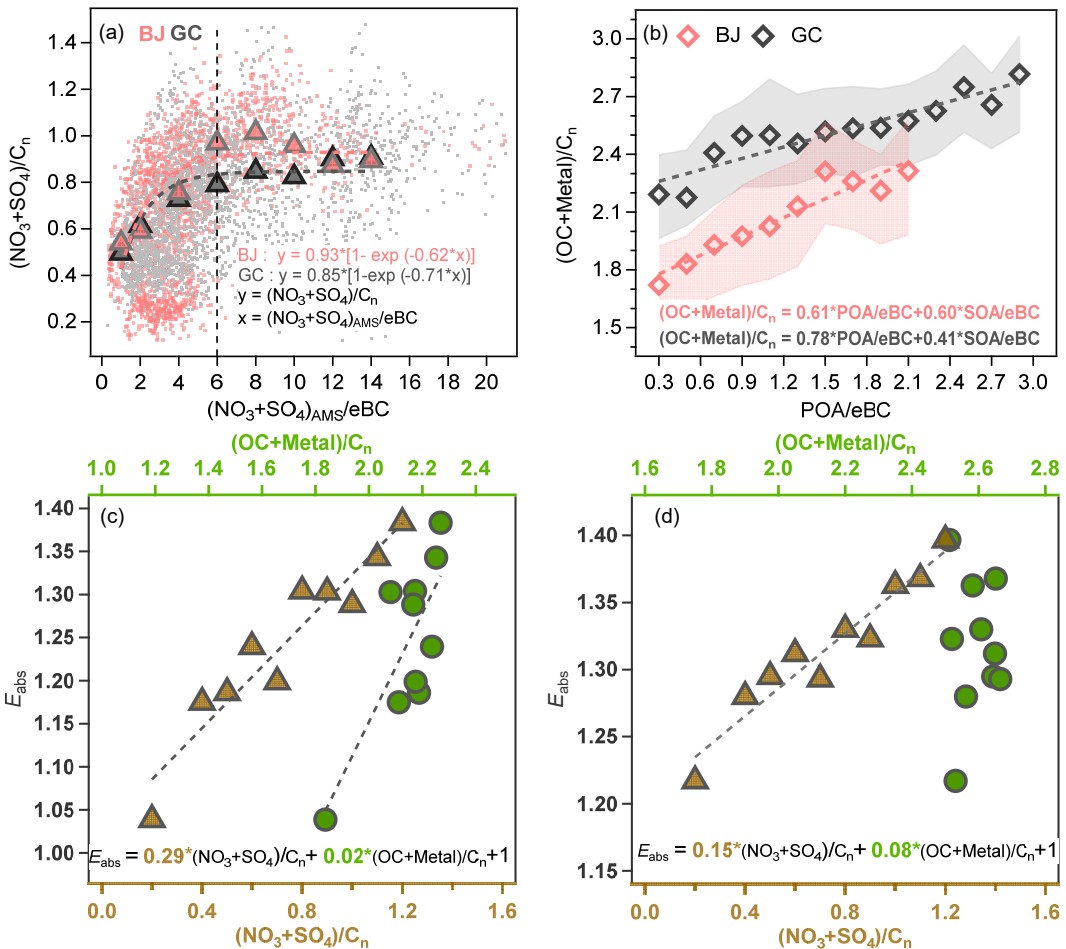


**Figure 4. Relationships between peak area ratios and mass concentration ratios (a) (NO₃ + SO₄)/Cₙ vs. (NO₃ + SO₄)ₐₘₛ/eBC and (b) (OC + Metal)/Cₙ vs. POA/eBC. Relationships between $E_{abs}$ and peak area ratios of coating materials (NO₃ + SO₄)/Cₙ and (OC + Metal)/Cₙ in (c) Beijing and (d) Gucheng. The triangles in (a) are the averages of different bins that are grouped according to (NO₃ + SO₄)ₐₘₛ/eBC. The rhombuses in (b) are the averages of different**

**bins that are grouped according to POA/eBC. The shades areas in (b) indicate 25th and 75th percentiles of the (OC + Metal)/Cₙ ratios. The triangles and circles in (c) and (d) are the averages of different bins that are grouped according to (NO₃ + SO₄)/Cₙ.**

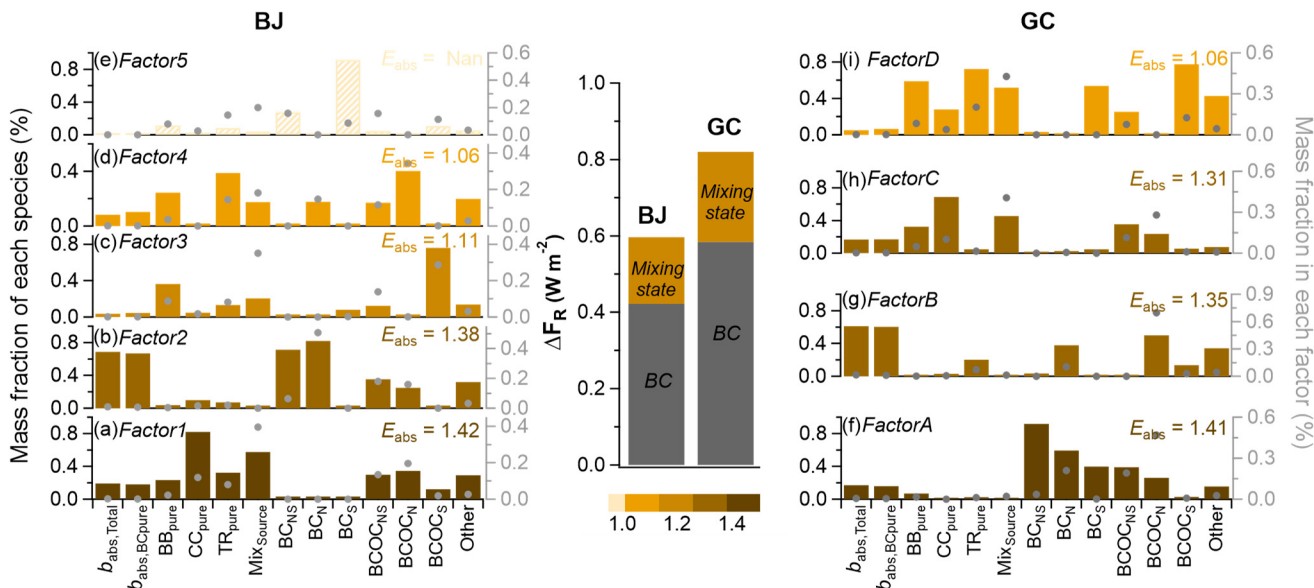

**Figure 5.** Factor profiles and their contributions to each factor identified by the positive matrix factorization model in Beijing and
Gucheng. $b_{abs, total}$ and $b_{abs, BCpure}$ represent light absorption (Mm$^{-1}$) of coated BC particles and pure BC, respectively. BC types e.g.,
$BB_{pure}$ in units of count.

