# Peer review of "Chemical composition and mixing state of BC-containing particles and the implications on light absorption enhancement"

_Atmospheric Chemistry and Physics, 2021_

## Author Comment (AC1)

**Response to Reviewers' comments**

We are thankful to the two reviewers for their thoughtful and constructive comments that help improve the manuscript substantially. We have revised the manuscript accordingly. Listed below is our point-to-point response in blue to each comment that was offered by the reviewers.

**Response to Reviewer #1**

The manuscript investigated the mixing state and light absorption enhancement of BC-containing particles at Beijing and Gucheng during winter. They found that coating of second inorganics has a larger enhancement than organic coatings. Overall, this study can improve the knowledge of BC light absorption enhancement and evaluation of BC light absorption during the atmospheric aging process. This can help reduce the uncertainties in BC climate effects. However, this manuscript is not well written. Many cases require additional information to clarify motivation, methodology, results, and interpretation. My impression of this paper is that it could be improved by considering the following suggestions. The revised paper should discuss these points, not just answer in the authors' response. I am willing to review the revised manuscript. Therefore, my recommendation for the editor is that this manuscript needs major revision.

**General comments:**

1. Captions of figures do not include all necessary information. Also, some figures are very confusing. Please see my specific comments below.

Following the reviewer's suggestion, we have expanded the descriptions of Fig. 3, Fig. 4, Fig. 5, and Fig. S2 according to specific comments.

2. In the Methods section, the authors could provide more information about sampling, analysis methods, instrumentation, etc. I understand some methods have already been published and are widely used in the literature (e.g., L125-127, "The detailed … in factor *i*."). However, it is better to provide short summaries of these methods in the main manuscript or SI. It is not clear how these methods work for me, and I have to go back to the original references.

Following the reviewer's suggestions, we added more information on measurements and analysis in the revised manuscript and supplementary.

In manuscript:

Please see our response to the specific comments 3, 4 and 5 below.

In supplementary:

"PMF analysis

PMF analysis was used to identify the effect of different mixing state on $E_{abs}$. The input of PMF analysis included $b_{abs, total}$, $b_{abs, BCpure}$ and 11 major types of BC-containing particles, and the uncertainties were determined by the following algorithms (Polissar et al., 1998; Petit et al., 2014) (Eq. 1):

$$
U_{ij} = \begin{cases} \frac{5}{6} * LOD_i & \text{if } C_j \leq LOD_i \\ \sqrt{u_i^2 C_i^2 + LOD_i^2} & \text{if } C_j \geq LOD_i \end{cases} \tag{1}
$$

$LOD_i$ represents the limit of detection and $u_i$ represents the relative uncertainty (in %) for each variable. The final uncertainty ($U_{ij}$) is determined by the LOD and $u$, which represents the $i$th species in $j$ row. LODs for the species were calculated as 3 times the standard deviation calculated during the clean period. After a careful evaluation of PMF solutions, five factors (Factor1, Factor2, Factor3, Factor4 and Factor5) in Beijing and four factors (FactorA, FactorB, FactorC and FactorD) in Gucheng were chosen to study the influence of different mixing state on $E_{abs}$."

3. Many discussions are not clear to me. Please see my specific comments.

Please see our responses to specific comments.

4. I did not see any discussion about uncertainties. Please add these like uncertainties in instruments and data.

Following the reviewer's suggestion, we added more discussions on uncertainties in the revised manuscript. The details are given in our responses to the specific comments of 2, 3, 4 and 17.

**Specific comments:**

1. In this manuscript, you used terms such as "coating" and "internally mixed". In this case, I assume you mean particles are internally mixed and core-shell morphology. However, it could also be partially coated or aggregated together, which you cannot tell from AMS or SP-AMS. Do you have any TEM images, EDX mapping, or shape factor measurement? If you do not have evidence to support core-shell morphology, I would like to see some discussions about the effects of different morphologies.

We thank the reviewer's comments. Unfortunately, we didn't have the measurements of TEM images, EDX mapping or shape factor in this study. The "coating" and "internally mixed" in our study referred to either core-shell or partially coated or aggregated together because they can be measured by single particle AMS (Bond and Bergstrom, 2006; Healy et al., 2012; Pratt and Prather, 2012; Bi et al., 2015; Chen et al., 2016; Xie et al., 2020). However, it is difficult to study the effects of different

morphologies because of the absence of TEM/EDX related measurements and the limitation of SPA-MS.

To clarify this, we added the description of these terms in Section 2.2 in the revised manuscript:
"According to previous studies, the coating materials on BC-containing particles measured by SPA-MS referred to chemical components that are partially or fully coated on BC (Bond and Bergstrom, 2006; Healy et al., 2012; Pratt and Prather, 2012; Bi et al., 2015; Chen et al., 2016; Xie et al., 2020)."

2.  I understand that you used 870 nm PAX to measure light absorption properties since many studies believe only BC can absorb at 870 nm. However, many studies have pointed out that brown carbon (BrC) can also absorb at 870 nm, leading to overestimating your eBC. Moreover, BrC can also scatter light at 870 nm, leading to overestimating your BC scattering properties. These two can result in different results of $\Delta F_R$. I did not see any discussion about these. Please add discussions.

We agree with the reviewer that brown carbon (BrC) can also absorb at 870 nm. As the reviewer mentioned that most previous studies showed that the total absorption at 870 nm was almost exclusively due to BC (Fialho et al., 2005; Sandradewi et al., 2008; Bahadur et al., 2012; Lack et al., 2014; Drinovec et al., 2015), and the BrC absorption accounted for only ~1% (Figure R1) (Yang et al., 2009). Therefore, the impacts of BrC on absorption and scattering at 870 nm are expected to be small. In the section of $\Delta F_R$ estimation, we assumed that the absorption measured by the TD-PAX at 870 nm during $T > 200°C$ was pure BC absorption. Then, we used the theoretical value of scattering properties of pure BC to estimate its radiative forcing in supplementary (Charlson et al., 1992; Chylek and Wong, 1995). This estimate only provided the theoretical effect of BC and the absorption enhancement caused by its coating on the radiative forcing and did not consider the scattering enhancement of BC due to the coatings.

Following the reviewer's suggestion, we expanded the discussions of the potential impact of BrC on radiative forcing estimation and clarified the description in the revised manuscript.

"It should be noted that BrC can also absorb light at 870 nm, leading to an overestimation of BC absorption. Considering that the contribution of BrC to the total absorption at 870 nm is typically small (< 1%) (Clarke et al., 2004; Fialho et al., 2005; Yang et al., 2009), the impact of BrC on the estimation of radiative forcing of BC is expected to be small."

"Based on $E_{abs}$ for each factor and its contribution to $b_{abs, BCpure}$, we estimated the direct radiative forcing ($\Delta F_R$) caused by pure BC at the top of the atmosphere (TOA) and the absorption $\Delta F_R$ enhanced by the mixed state of BC-containing particles (Chylek and Wong, 1995; Chen and Bond, 2010)."

[Figure]

Figure R1. Apportioning of total light absorption to black carbon, brown carbon, and dust: project medians of their absolute absorption (left panel) and relative contributions (right panel) (Yang et al., 2009).

3. In this study, the max temperature of the thermodenuder (TD) is 250°C. However, this temperature might not remove all BrC and inorganics (see *"Two-stage aerosol formation in low-temperature combustion"* and *"The Brown-Black Continuum of Light-Absorbing Combustion Aerosols")*. How do you account for that? Also, did you count particle loss in the TD?

Thank the reviewer for pointing this out. The highest temperature *(T)* of 250°C indeed could not evaporate all chemical components coated on BC which could lead to an underestimation of $E_{abs}$ because of overestimated absorption. However, such an underestimation can be relatively offset by the TD loss in our study. The loss of BC particles in TD was experimentally determined by using the pure BC particles within different temperature (Figure R2). The results showed that the BC particle mass loss was about 14% in TD line at $T > 200$°C. In addition, we found that residual material accounted for ~4% of the total NR-PM$_1$ after heating and volatilization at $T>200$°C according to our previous studies in Beijing and Gucheng (Figure R3) (Xu et al., 2021). The residual NR-PM$_1$ accounted for ~50% of the total residual mass, i.e., NR-PM$_1$ + BC. This is consistent with pervious study showing that the coating material evaporated below 140°C contributed to ~80% of the absorption enhancement induced by coatings (Ma et al., 2020), and $b_{abs}$ decreased by ~4% after the evaporation of sulfate and semi-volatile organic matter between 140 and 210°C (Ma et al., 2020). As a result, the residual materials led to an overestimation of the $E_{abs}$ by ~16% at 210°C. Considering the underestimation of ~14% of $E_{abs}$ due to the TD loss and the overestimation of ~16% due to the coated materials at 210°C, we did not do TD loss correction to reduce the

uncertainties caused by the lensing effect.

We thank the reviewer's comments and have added an explanation in the revised manuscript and Figure R2 in supplementary:
"Note that we did not do TD loss correction in this study. The reason is that the underestimated $E_{abs}$ of ~14% due to the TD loss (Fig. S1) can be relatively offset by the overestimation of ~16% due to the coating materials that did not evaporate at 210°C (Ma et al., 2020; Xu et al., 2021)."

[Figure]

Figure R2. The loss of pure BC particles in TD within different temperature.

[Figure]

Figure R3. Thermograms of non-refractory submicron aerosol species including organics (Org), sulfate (SO₄), nitrate (NO₃), ammonium (NH₄) and chloride (Chl). The mass fractions of size-resolved non-refractory submicron aerosol (NR-PM₁) species as a function of TD temperature are also shown (Xu et al., 2021).

4. In your calculation of absorption enhancement ($E_{abs}=b_{abs,total}/b_{abs,BC}$), $b_{abs}$ can be different for the same component if their concentration is different before and after TD. It is better to use MAC or imaginary part of the refractive index.

We agree with the reviewer that MAC or imaginary part of the refractive index method could be used to determine the $E_{abs}$. In addition, the TD-method has been widely used in previous studies and the results have been shown to be valid (Cappa et al., 2012; Nakayama et al., 2014; Healy et al., 2015; Zhang et al., 2016). In our study, to minimize the uncertainty due to the variation of $b_{abs}$ mentioned by the reviewer, the absorption coefficients of ambient particles during the periods of TD measurements were obtained by interpolation from adjacent measured ambient $b_{abs}$, and the $E_{abs}$ was then calculated as the ratio of $b_{abs, totalInterp}$ to $b_{abs, BCpure}$ instead of using the ratio of $b_{abs, total}$ to $b_{abs, BCpure}$ within two adjacent 15 minutes. In addition, we continuously measured the $b_{abs}$ of ambient atmosphere by PAX for about 6 days to evaluate the uncertainty of interpolation method. Figure R4 shows the comparisons of the interpolated and the measured data. We could find the good consistency between the results. Furthermore, the relative deviation by interpolation was about 6% compared with the measured data. These results indicate that the uncertainty of experiment was small and the results were reliable. And this uncertainty could be minimized in relative long-term observations.

To clarify this, we added a description about the determination of $E_{abs}$ in the revised manuscript as follows:
"To minimize the uncertainties due to the changes during the measurements of TD and bypass, the $b_{abs, total}$ was obtained from the linear interpolation of measured ambient absorptions adjacent to the TD time, and the $E_{abs}$ was then determined as the ratio of $b_{abs, total}$ to $b_{abs, BCpure}$ that was determined as the thermodenuded particle absorption in TD line at $T > 200°C$."

[Figure]

Figure R4. A comparison of interpolated and measured $b_{abs}$.

5. L114-118, "Seven particles … are shown in Table S1." How many signals of each fragment do you need to classify a particle to a type? How did you decide the thresholds?

In our study, the methods used to obtain particle types were based on previous studies on single particle measurements. In order to show more details mentioned by the reviewers, we have enriched Table S1 with the references, thresholds and the definition of particle types in revised supplementary as follows (Table R1):

Table R1. A summary of abberviations and descrptions of BC-containing particle types.

| Description of type or species | Typical ions | Abbreviation | References |
|---|---|---|---|
| BC only from biomass burning | $39K^+$ (peak area >1500) and two of the signals in $45[CHO_2]^-$, $59[C_2H_3O_2]^-$ and $73[C_3H_5O_2]^-$ (peak area >200). | $BB_{pure}$ | (Silva et al., 1999; Healy et al., 2010) |
| BC only from coal combustion | $7Li^+$ (peak area >200) or $23[Na]^+$, $27[Al]^+$, $43[AlO]^-$ (peak area >200) or $80[SO_3]^-$, $97[HSO_4]^-$ (relative peak area >2%). | $CC_{pure}$ | (Zhang et al., 2009; Healy et al., 2010) |
| BC only from traffic emission | $55[Mn]^+$ (peak area >200 without $Na^+$ and $Al^+$) or $40[Ca]^+$ (with abundant nitrate) or $79[PO3]^-$ (with abundant nitrate) or $51[V]^+$ and $67[VO]^+$ (peak area >200). | $TR_{pure}$ | (Yang et al., 2017) |
| BC internally mixed more than one sources | Same as the above. | MixSource | |
| BC internally mixed with nitrate | $46[NO_2]^-$ and $62[NO_3]^-$ (relative peak area >70%). | $BC_N$ | The selected conditions about nitrate, sulfate and OC (Dall et al., 2012; Sierau et al., 2014; Chen et al., 2016; Zhou et al., 2016; Arndt et al., 2017; Cheng et al., 2017; Zhang et al., 2019). |
| BC internally mixed with sulfate | $97[HSO_4]^-$ (relative peak area >70%). | $BC_S$ | |
| BC internally mixed with nitrate and sulfate | $46[NO_2]^-$ and $62[NO_3]^-$ are comparable with $97[HSO_4]^-$. | $BC_{NS}$ | |
| BC internally mixed with OC and nitrate | three of the signals in $37[C_3H]^+$, $43[C_2H_3O]^+$, $51[C_4H_3]^+$ and $63[C_5H_3]^+$ (relative peak area >2%) with nitrate. | $BCOC_N$ | |
| BC internally mixed with OC and sulfate | three of the signals in $37[C_3H]^+$, $43[C_2H_3O]^+$, $51[C_4H_3]^+$ and $63[C_5H_3]^+$ (relative peak area >2%) with sulfate. | $BCOC_S$ | |
| BC internally mixed with OC, nitrate, and sulfate | three of the signals in $37[C_3H]^+$, $43[C_2H_3O]^+$, $51[C_4H_3]^+$ and $63[C_5H_3]^+$ (relative peak area >2%) with comparable nitrate and sulfate. | $BCOC_{NS}$ | |

6. L130-131, "BC-containing … (Xie et al., 2020)." I am curious why there is a discrepancy between your study and Xie et al.?

This discrepancy was likely due to the higher mass fraction of eBC in the total mass concentration in our study than that observed by Xie et al. (2020) (9.3% vs. 6.1%). In addition, the new version of HR-SPAMS showed higher performance than SPA-MS, which improved the sensitivity of low intensity signal greatly and reduced the probability of missing BC fragments with low signals (Shen et al., 2018; Zhong et al.,

2021). In the revised manuscript, we added an explanation as below:

"The BC-containing particles accounted for 62% of the total particles in Beijing, lower than that in Gucheng (73%) yet higher than that in winter Beijing 2018 (55%) (Xie et al., 2020) likely due to the higher mass fraction of eBC in this study (9.3% vs. 6.1%)."

7. L135-137, "Comparatively, … at the urban site." This is true for large particles that $SO_4$ is generated from fog or cloud processing. $SO_4$ could be generated from anthropogenic sources such as coal combustion for small particles. Do you have any measurements to show that these $SO_4$ are coming from the aging processes?

We agree with the reviewer that $SO_4$ could be directly emitted from coal combustion. In fact, $SO_4$ in $BCOC_S$ was more likely from the relatively fresh coal combustion emissions, while $SO_4$ in $BCOC_{NS}$ was more likely from the aging processes (Figure R5) in this study. However, it is very challenging to separate the sulfate from direct emissions and secondary formation. In general, the sulfate in the atmosphere was dominantly from gas-phase and aqueous-phase oxidation of $SO_2$. Therefore, sulfate has been widely used an indicator of regional processes. Besides, the typical fragment of secondary organic aerosol, i.e.,$43[C_2H_3O]^+$ (Healy et al., 2010; Chen et al., 2019) showed a much higher relative area in BC-containing particles in BJ than GC (Figure 1). These results also indicated that BC particles were more aged at the urban site. We changed the description in the revised manuscript as below:

"Comparatively, more than 80% of BC-containing particles were internally mixed with $SO_4$ in Beijing while those in Gucheng accounted for less than 60%. Considering the higher relative area of secondary organic fragment of $43[C_2H_3O]^+$ (Healy et al., 2010; Chen et al., 2019) in BJ, we concluded that BC particles were likely more aged at the urban site. Another support is much lower primary emissions of biomass burning and coal combustion in BJ than GC (Sun et al., 2020b)."

[Figure]

Figure R5. Relationship between the mass fraction of OA factors and the number fraction of BC-containing types.

8. Section 3.2 is not clear to me. I might misunderstand some concepts. Please clarify my following questions:
1. L163-165, "Moreover, the number … in urban region." Why do you say this? The correlation between increased $BC_N$ concentration and their role in pollution formation is not clear to me. First, do you have any particle concentration and air quality measurements to show these days are highly polluted? Second, what important roles are you mentioned here? I can see the potential correlation between $BC_N$ formation and RH, but you need to explain a little more between $BC_N$ and urban region pollution. Also, I suggest you do some statistic analysis to show correlations. Similar to any comparison you did in the manuscript.

We thank the reviewer for pointing this out. We did not show the mass concentration of PM in the original manuscript. Following the reviewer's comments, we revised the figure. As shown in Figure R6, the mass concentrations of PM (NR-PM$_1$ + eBC) increased substantially as a function of RH in both BJ and GC except the period of RH = 90–100% which was likely due to the scavenging effect of precipitation. Following the reviewer's suggestion, we did the correlation analysis (Figure R7) which also supported that the $BC_N$ played an important role in the formation of severe pollution in urban region. We changed the description in the manuscript and added the Figure R7 as Figure S4 (a) in the revised supplementary.

"Moreover, the number fraction of $BC_N$ increased gradually as the increase of RH and dominated BC particles (25–30%) at RH = 70–100% in BJ. Considering the similar increases of PM (NR-PM$_1$ + eBC) as a function of RH (Figs. 3a and S4), the RH dependence of $BC_N$ suggested that the newly formed nitrate that coated on fresh BC

played an important role in the formation of severe pollution in urban region."

[Figure]

Figure R6. Variations of number fractions of BC-containing particle types and the sum of mass concentration of NR-PM$_1$ and eBC with RH in (a) Beijing and (b) Gucheng. The bottom and top error bars represent 25th and 75th percentiles.

[Figure]

Figure R7. The relationship between the mass concentration of PM (= NR-PM$_1$ + eBC) with the number fraction of BC$_N$ in BJ.

2. L166-167, "This result … in rural area." See comment 8(a).

We have corrected Figure 3(b) and changed it to Figure R6(b) (see comment 8(1)). We also performed the correlation analysis of the NR-PM$_1$+eBC versus the number fraction of BCOC$_N$ in GC to support our conclusion (Figure R8) which was included in the revised supplementary.

[Figure]

Figure R8. The relationship between the mass concentration of PM (= NR-PM$_1$ + eBC) and the number fraction of BCOC$_N$ in GC.

3. L166-171, "In addition, we … zhang et al./. 2021)." I do not understand this. Higher RH has lower BCOC$_N$. Does that mean more BCOC$_N$ formation depends on photochemical? Moreover, how could sulfate formation not affect BC mixing state and light-absorption? This does not make sense to me.

In this study, we found that higher RH was associated with lower BCOC$_{NS}$, not the BCOC$_N$ ("the fraction of BCOC$_{NS}$ decreased obviously as a function of RH in GC"). We found that BCOC$_{NS}$ formation was more affected by the photochemical than aqueous-phase reaction in GC. Another support is both the number count and fraction of BCOC$_{NS}$ increased significantly during daytime (Figure R9 in supplementary). As RH was above 70% (Figure R10) (Sun et al., 2021), $E_{abs}$ was fairly stable in GC despite the increased sulfate contribution. Note that the number fraction of BC type mixed with SO$_4$ did not increase with the increases of RH in GC although the mass fraction of SO$_4$ in PM increased significantly. Therefore, we inferred that only a relatively small fraction of sulfate formed through the aqueous-phase chemical reaction was mixed with BC, which had a minor effect on the overall BC mixing state and absorption enhancement. To make it clear, we added more analysis in the revised manuscript.

"Considering the increased SO$_4$ mass fraction yet the relatively stable $E_{abs}$ at RH > 70% (Sun et al., 2021), we inferred that aqueous-phase formation of sulfate at high RH level appeared not to affect BC mixing state substantially, consistent with previous study (Zhang et al., 2021)."

[Figure]

Figure R9. Diurnal variation of $BCOC_{NS}$ in GC.

[Figure]

Figure R10. Changes of (a) $E_{abs}$ and mass fractions of aerosol species as a function of RH in (b) Beijing and (c) Gucheng (Sun et al., 2021).

4. In figure 3, why did you name the earlier case as case 2? How do you define P0-P5?

The two haze events showed similar formation processes. The formation processes of P0, P1 and P2 in Case1 were characterized by obvious increases in NR-PM$_1$ and

significant changes in the number fraction of most BC types. Similar periods in Case2 were named as P3, P4 and P5 to reduce confusions in the description. During P0, P1 and P2 of case1, the freshly emitted BC was gradually mixed with nitrate and OC causing high $E_{abs}$, similar to P3 and P4. However, P5 showed a decrease in $BCOC_N$ number fraction due to inconsistencies in the primary emissions (BB and FF), which did not occur in Case1. To make the manuscript more logical, we named the earlier case as case 2.

5. L175, "As a consequence, … in half-day." You showed that $E_{abs}$ increased by decreasing $BCOC_S$ and increasing $BCOC_N$. Could that be because $BCOC_S$ is less volatile than $BCOC_N$, so that after TD, more $BCOC_N$ (coating) can be removed?

Thank the reviewer's comments. The single particle measurements did not use the TD in this study. We agree with the reviewer that $BCOC_S$ is expected to be less volatile than $BCOC_N$ because more volatile properties of nitrate than sulfate.

6. L180-187, "As shown in Fig. 3d, … at high PM level." First, how do you conclude the mountain valley winds influenced the concentration of the fine particles? Do you have any evidence to show the wind direction, or have any references explained that before? Moreover, since you mentioned the fine particles were dominated by fossil fuel OA, which indicates local traffic emission based on my knowledge. Then, should the diurnal variations cause by the changes in traffic conditions? Second, you did not show the diurnal cycle of $E_{abs}$. Moreover, I also do not understand the effect of mountain valley winds since you did not clearly explain that. Furthermore, could the variation in Eabs be due to changes in chemical composition? Third, you mentioned that after P5, BBOA was stable, and FFOA increased. However, I found BBOA decreased, and FFOA increased until midnight of 11/10, then decreased. I do not see you providing BBOA and FFOA concentration at other times. It will be interesting to see that. Last, you said the $BCOC_N$ and BC were high in the last sentence. I am not sure what the period is. Could you clarify that? Moreover, I also do not understand how you conclude that fresh fossil fuel BC might mix with OC and nitrate at a high PM level.

The mountain valley wind is a common phenomenon in Beijing (Sun et al., 2016). The wind direction often changes from the north-northwest to the south after 12:00–14:00, and then brings the air pollutants in the south to Beijing, leading to the increases in mass concentrations of most aerosol species. After midnight, the wind direction changes from the south to the north-northwest and brings the clean air from the mountain area, leading to the decreases in air pollutants. Such impacts of mountain valley winds on air pollution in Beijing have been reported in previous studies (Chen et al., 2015; Sun et al., 2016). The similar impact was also observed in this study. As shown in Figure R11, the wind direction switched to the south after 13:00, and the mass concentration started to increase, while after midnight, the wind direction changed to the north again, and the mass concentration decreased again.

The FFOA in our study comprised organic aerosol from coal combustion and traffic

emissions that cannot be separated by PMF analysis. The mass concentration of CCOA is generally much higher than HOA according to previous measurements in cold season in Beijing (Xu et al., 2019; Li et al., 2021). Considering that coal combustion has been prohibited in entire Beijing and the high concentration CCOA is expected to be mainly from surrounding regions (Sun et al., 2021). The diurnal variation of traffic emission in Beijing shows high values of HOA during 23:00–6:00 because only heavy-duty vehicles and diesel trucks are allowed to enter the city during this time (Xie et al., 2019) which was not consistent with the changes in FFOA in this case. Besides, the diurnal variation of $E_{abs}$ mentioned in the manuscript in this part was also elaborated about Case2. We did not show the average diurnal cycle of $E_{abs}$ because it can be seen in the time series of $E_{abs}$ in Figure 3d, and we found the similar variations between FFOA and $E_{abs}$ (Figure R13d). Therefore, the variations in $E_{abs}$ were likely caused by the different chemical composition and mixing state because of the changes in air masses due to the impact of mountain valley winds.

As the reviewer mentioned in the third point, after P5, BBOA decreased and FFOA increased until midnight of 10 November, then decreased. The "after P5" we described in the manuscript means from 21:00 on 9 November to 0:00 on 10 November (P6 in Figure R12). We used the term "relatively stable" to describe BBOA because the variation of BBOA mass concentration was below 1.5 μg m$^{-3}$ during P6. And the time for the increased proportion of BCOC$_N$ in BC also occurred during the same period (P6). Considering the consistent variation with FFOA, we concluded that BCOC$_N$ was likely from fossil fuel combustion (likely coal combustion) because BBOA (another major source of BC) was relatively stable. Similar to P6, during 14:00 - 19:00 on 8 November (Figure R13d), BCOC$_N$ also increased with the increase of FFOA while BBOA was stable. Therefore, we concluded that BC emitted from fossil fuel emission was likely mixed with OC and nitrate at high PM levels

Following the reviewer's comments, we clarified the timing of Figure 3d, added wind speed and wind direction in Figures 3c and 3d (Figure R13), added the Figure R11 in supplement, and changed the statement in revised manuscript as below:

"As shown in Figs. 3d and S5, the sources of fine particles were dominated by fossil fuel OA and presented strong diurnal variations consistent with changes in wind direction due to the influences of mountain valley winds (Sun et al., 2016). As a result, $E_{abs}$ also presented a relatively consistent variation with FFOA characterized by the higher values during nighttime, indicating that the different chemical composition and mixing state because of the changes in air masses caused by the mountain valley winds had affected the light absorption enhancement of BC (Ding et al., 2021)."

"After the P5 period (from 21:00 on 9 November to 0:00 on 10 November), the BBOA decreased slightly and FFOA increased significantly. The proportion of BCOC$_N$ in BC was correspondingly higher. Similar variation could also be found during 14:00 - 19:00 on 8 November. These results indicated that BC emitted from fossil fuel emission was likely mixed with OC and nitrate at high PM level."

[Figure]

Figure R11. Time series of wind speed (WS), wind direction (WD), PM, FFOA and BBOA in Case2.

[Figure]

Figure R12. Temporal variations of $E_{abs}$, number fractions of BC-containing particle types and mass concentration of species during P5 and P6 in Case2.

[Figure]

Figure R13. Variations of number fractions of BC-containing particle types with RH in (a) Beijing and (b) Gucheng. Temporal variations of $E_{abs}$, number fractions of BC-containing particle types and mass concentration of species during polluted (c) Case1 and (d) Case2. The bottom and top error bars represent 25th and 75th percentiles.

7. L190-191, "Such differences were ... during daytime." Have you considered the heterogenous nitro reactions during nighttime?

Thank the reviewer's comments. The heterogeneous reaction could play a role in nitrate formation at nighttime. However, considering the low $O_3$ concentration in winter severe haze episodes, the formation of $N_2O_5$, and subsequent hydrolysis to form nitric acid could not be important. Indeed, previous studies have found that gas-phase oxidation of $NO_2$ is far more important than nighttime $N_2O_5$ reaction in nitrate formation in winter in Beijing (Chen et al., 2020). In addition, the nitrate formed at nighttime should be mixed with both $BC_N$ and $BCOC_N$, leading to the increases in the number fraction of both $BC_N$ and $BCOC_N$. However, the increase of $BC_N$ number fraction at nighttime was not observed in GC in our study (Figure R14). Therefore, we concluded that nighttime heterogenous reaction of nitrate unlikely played an important role.

[Figure]

Figure R14. Diurnal variation of number counts and number fraction of $BC_N$ in GC.

9. L198, should it be Fig. S1 instead of Fig. S2? Also, Fig S2 is very confusing. What are the solid line and dashed line?

Yes, we misspelled Fig. S1 as Fig. S2 and corrected it in the revised version. We revised Fig. S2 in the new version of the manuscript.

[Figure]

Figure R15. Diurnal variations of six types of BC-containing particles in Beijing and Gucheng.

10. Figure 4: what are these triangles in the figures? For figures 4 a and b, I suggest switching the x and y axis since the x axis depends on the Eabs, which depends on the $(NO_3+SO_4)/C_n$ and $(OC+Metal)/C_n$.

Thank the reviewer's comments. The triangles are bin averages that are grouped

according to x-variables. We revised the figure caption in the revised manuscript.

"Figure 4. Relationships between peak area ratios and mass concentration ratios (a) $(NO_3 + SO_4)/C_n$ vs. $(NO_3 + SO_4)_{AMS}/eBC$ and (b) $(OC + Metal)/C_n$ vs. POA/eBC. Relationships between $E_{abs}$ and peak area ratios of coating materials $(NO_3 + SO_4)/C_n$ and $(OC + Metal)/C_n$ in (c) Beijing and (d) Gucheng. The triangles in (a) are the averages of different bins that are grouped according to $(NO_3 + SO_4)_{AMS}/eBC$. The rhombuses in (b) are the averages of different bins that are grouped according to POA/eBC. The shades areas in (b) indicate 25th and 75th percentiles of $(OC + Metal)/C_n$ ratios. The triangles and circles in (c) and (d) are the averages of different bins that are grouped according to $(NO_3 + SO_4)/C_n$."

We thank the reviewer's suggestion for switching the x and y axis. In our study, the measurement of AMS is aerosol mass concentration while that of SPA-MS was the number concentration of chemical components mixed with BC. In this part of the analysis, we wanted to investigate how the mixing efficiency of atmospheric chemical components with BC changed as the changes in mass concentrations. We therefore kept the axis in the manuscript.

11. L208-209, "These results indicated … exceeded ~6." This is not clear to me. Can you explain a little bit more? You are not showing a time series of these two ratios. Does eBC increase with aging time? Also, I do not know how you calculate $(NO_3+SO_4)_{AMS}/eBC$. Based on your context, it seems like this ratio will keep increasing due to aging (although you did not provide the time evolution of that ratio). Then if that keeps increasing, why will the $(NO_3+SO_4)/Cn$ ratio stop increasing? I guess that eBC is initially increased due to increasing $NO_3$ and $SO_4$, which have higher $E_{abs}$ (eBC=$b_{abs}$/MAC=$E_{abs}*b_{abs\_BC}$/MAC). $NO_3+SO_4$ increased at the beginning and then reached a steady state. Then something happened, which decreased the overall $E_{abs}$ (maybe coating of organics?). Anyway, I might be wrong. However, please show me more evidence.

Previous studies found that the coating materials of black carbon and their resulting absorption enhancements did not necessarily increase with pollution and aging processes in the ambient atmosphere (Cappa et al., 2019; Liu et al., 2019a; Sun et al., 2020a), while it depended more on the mixing efficiency of secondary components with BC. For example, we found the aqueous-phase formation of sulfate at high RH level appeared not to affect BC mixing state substantially.

The analysis in this part was based on the total measurements in BJ or GC to explore the changes in the mixing efficiency of secondary inorganic components with BC during the aging processes in urban or rural ambient sites. Different from the case studies with a detailed time series, we presented the total data set as dots and the bin results as markers in the graph for clarification. The $(NO_3+SO_4)_{AMS}/eBC$ was determined as the ratio of the sum of $NO_3$ and $SO_4$ mass concentration measured by AMS to eBC mass concentration measured by PAX in the ambient atmosphere. As

shown in right panel in Figure R16 right panel, the simultaneous growth of $(NO_3+SO_4)_{AMS}/eBC$ and NR-PM$_1$ indicates that the increase of $(NO_3+SO_4)_{AMS}/eBC$ can indicate the process of increasing pollution, and the maximum ratio of $(NO_3+SO_4)_{AMS}/eBC$ was ~14. The eBC mass concentration varied relatively small (the variation below 2 µg m$^{-3}$) with the increase of $(NO_3+SO_4)_{AMS}/eBC$ at both sites (left panel in Figure R16), indicating that the increases in $(NO_3+SO_4)_{AMS}/eBC$ were mainly due to the stronger secondary formation during the aging process.

As shown in Figure 4(a) in the manuscript, the peak area ratio of $(NO_3 + SO_4)/C_n$ gradually became stable at both sites as the increase of $(NO_3 + SO_4)_{AMS}/eBC$. This was mainly due to the fact that, when the NR-PM$_1$ in the ambient atmosphere was below ~130 µg m$^{-3}$ based on the measurements in BJ and GC, the mixing efficiency of NO$_3$ and SO$_4$ with BC reached a maximum of $(NO_3 + SO_4)_{AMS}/eBC$ = ~6, and the additional NO$_3$ and SO$_4$ were not mixed with BC. Following the reviewer's comments, we revised the expression in the manuscript to make it clearer.

"These results indicated that BC was rapidly aged and internally mixed with secondary inorganic components during the early stage of haze episode, and appeared to be fully aged when the mixing efficiency of NO$_3$ and SO$_4$ with BC reached a maximum, i.e., $(NO_3 + SO_4)_{AMS}/eBC$ = ~6."

[Figure]

Figure R16. Relationship of eBC and NR-PM$_1$ with $(NO_3+SO_4)_{AMS}/eBC$ in BJ and GC.

12. L209-211, "Different from … at both sites." What does this tell you? What is your purpose in comparing these two variables, and why do you not use the same format as Fig. 4a?

Based on the tight correlation between $(OC + Metal)/C_n$ and POA/eBC, we found that the POA co-emitted with BC was more easily internally mixed with BC than secondary inorganic components such as SO$_4$ and NO$_3$ although the mass concentration of POA was much lower than that of SIA. In this section, we would like to discuss the response of OC coated on BC to the increase in atmospheric primary composition. We expanded the descriptions in the revised manuscript.

"These results indicated that the POA co-emitted with BC was more easily internally mixed with BC than secondary inorganic components (e.g., $SO_4$ and $NO_3$) although the mass concentration was much lower than SIA at both urban and rural sites."

The statistical analysis showed that the linear correlation coefficient ($R^2$) of ($NO_3$ + $SO_4$)/$C_n$ vs. ($NO_3$+$SO_4$)$_{AMS}$/eBC was 0.22 and 0.23 in BJ and GC, respectively. However, the exponential fit presented high correlation coefficient of ($NO_3$ + $SO_4$)/$C_n$ vs. ($NO_3$+$SO_4$)$_{AMS}$/eBC (0.99 and 0.98 in BJ and GC, respectively). The correlation coefficient is calculated as the ratio of the regression sum of squares to the total sum of squares. In addition, the mass concentration ratios of (OC + metal)/$C_n$ and POA showed a significant linear correlation, supported by the high linear correlation coefficients in BJ and GC (0.91 vs. 0.86). Thus, we used the different fit methods in Figs. 4a and 4b.

13. L211-212, "Moreover, … ($R^2$=0.95)". Where are these results presented?

The correlations between (OC + Metal)/$C_n$ and SOA/eBC in BJ and GC are shown in Figure R17, which have been added as Figure S7 in the revised supplementary.

[Figure]

Figure R17. Correlation between peak area ratio of (OC + Metal)/$C_n$ and mass concentration ratio of SOA/eBC (SOA = LO-OOA + MO-OOA in BJ and = OOA + aq-OOA in GC).

14. L223-225, "These results … at the urban site." Did you compare mass spectra of TD and bypass? Could that be because OC and metals are less volatile, they were not removed by TD efficiently?

Thank the reviewer's comments. It is a very good point. Unfortunately, the TD was not coupled with SPA-MS in this study, and we were unable to compare the single particle mass spectra between the TD and BP line. Like the reviewer mentioned, it could be due to the OC and metals that did not evaporate completely in TD. In fact, we evaluated the underestimation of $E_{abs}$ caused by the residual material after heating and volatilization at $T$>200°C and found that it was likely offset by the overestimation of $E_{abs}$ due to TD loss (see the response to specific comment 3). Therefore, the removal efficiency TD

unlikely affected $E_{abs}$ significantly.

We clarified the deployment of SPA-MS in the revised manuscript.

"A high-resolution single particle aerosol mass spectrometer (HR-SPAMS, Hexin Analytical Instrument Co., Ltd.) was deployed independently in the same room for measuring the mixing state and chemical composition of single particles in ambient atmosphere."

"The sources of bulk OA from HR-AMS measurements were analyzed by positive matrix factorization (PMF) and five OA factors were identified at both urban and rural sites, including…"

15. L227-229, "Combined with … at the rural site." How do you find that? Do you have any TEM images or EDX mapping? Mass spectra only can tell you these species exist in the same particle. Based on mass spectra, you cannot get morphology (partially coating or aggregate or core-shell).

We agree with the reviewer. In the absence of TEM or EDX measurements, we can only infer the conclusions based on literature researches and data analysis. Previous studies found that, $E_{abs}$ is ~1 when the non-BC material is partially filling in the voids between BC spherules or attached to them (Moteki et al., 2014; Liu et al., 2017). As shown in Figure R18 (Liu et al., 2017), $E_{abs}$ is >1 when the non-BC material is sufficient to encapsulate the BC. In our study in GC, when the secondary formation is weak (($NO_3$ + $SO_4$)/$C_n$~0), $E_{abs}$ showed a high value of ~1.2. Considering the ratio of (OC + Metal)/$C_n$ is ~2.5, we further inferred that OC and metals were not only as filler materials but also coated on fresh BC and induced light absorption enhancement at the rural site. We have added the description in the revised manuscript

"Pervious study showed that $E_{abs}$ is >1 when the non-BC material is sufficient to encapsulate the BC (Liu et al., 2017). Considering the peak area ratio of (OC + Metal)/$C_n$ was ~2.5, we inferred that OC and metals were not only as filler materials but also likely coated on fresh BC and induced light absorption enhancement at the rural site."

[Figure]

ARTICLES

NATURE GEOSCIENCE DOI: 10.1038/NGEO2901

**Figure 1 | Measured and modelled optical properties as a function of mass ratio. a,** Particle number distribution against $M_R$ under different environments. **b,** Single-particle scattering cross-section (at 2 fg mass) derived from various optical models and from direct observation, relative to the core–shell model. Each point represents the median value of single-particle data in each $M_R$ bin. The bars represent the variation from varying non-BC density between 1.0 and 1.2 g cm$^{-3}$ (BN and SF) and 0.9–1.1 g cm$^{-3}$ (CI). The grey bar shows the range of $M_R$ where transition occurs. TD, thermo-denuder.

Figure R18 (Liu et al., 2017).

16. Equation 1-6: How did you get these equations? Why are there so many variables? What is the difference between 1 and 2, 3 and 4, 5 and 6? You did not provide any explanations for these equations.

Thank the reviewer for pointing this out. The eqs.1–4 are used to explore the mixing efficiency of chemical components with BC at urban and rural sites, and the Eqs. 5–6 are used to discuss the effect of BC coating material on $E_{abs}$ at different measurement sites. They are obtained from the linear or exponential fits of Figure 4. Following the reviewer's comments, we deleted these equations and stated as below to avoid confusions in the revised manuscript.

"Moreover, we predicted the $E_{abs}$ using the statistical equations in Fig. 4 with aerosol species measured by AMS."

17. L232-233, "The predicted … (Fig. S4)." Have you done any statistical analysis to show the correlation between predicted and measured $E_{abs}$?

The average measured and estimated $E_{abs}$ values were 1.21 (±0.12) and 1.22 (±0.18), respectively in BJ, and 1.31 (±0.15) and 1.25 (±0.07), respectively in GC. As shown in Figure R6, the correlation coefficient between measured and predicted $E_{abs}$ was 0.63

and 0.51 in BJ and GC, respectively. In addition, we assessed the uncertainty of the estimated $E_{abs}$ by calculating the absolute ratio of ($E_{abs, measured}$ - $E_{abs, estimated}$) to $E_{abs, measured}$ (Figure R19). The results showed that the uncertainty of the estimated $E_{abs}$ in both BJ and GC was below 10% for more than 75% of the estimates.

We thank the reviewer's comments, and added Figure R19 as Figure S8 in the revised supplementary. Also, we expanded the description in revised manuscript.

"The estimated $E_{abs}$ showed overall agreements with the measured values in both BJ and GC. Although the correlation was not significant (Fig. S8), the average measured and estimated $E_{abs}$ values were similar which are 1.21 (±0.12) and 1.22 (±0.18), respectively in BJ, and 1.31 (±0.15) and 1.25 (±0.07), respectively in GC. Also, the uncertainty estimated from the difference of measured and predicted was overall below 10% in both BJ and GC indicating that the approach is reasonably well to estimate $E_{abs}$."

[Figure]

Figure R19. Comparison between measured and estimated $E_{abs}$ in BJ and GC. The uncertainty is determined by the absolute ratio of ($E_{abs, measured}$ - $E_{abs, estimated}$) to $E_{abs, measured}$.

18. L239-240, "Note that … Factor5 in BJ." It is not clear which factor is which. Please clarify that either here or in the Methods section.

Following the reviewer's suggestion, we revised Figure 5 which was shown as Figure R20, and cited it more precisely in the revised manuscript. For instance, "Note that $E_{abs}$ was not estimated when the factor contributed negligibly to the total BC, such as Factor5 in BJ." was changed to "Note that $E_{abs}$ was not estimated when the factor contributed negligibly to the total BC, such as Factor5 in BJ (Fig. 5e)."

"Five and four factors were identified in BJ and GC, respectively to elaborate the influence of different mixing state on $E_{abs}$ (Fig. 5)."

[Figure]

Figure R20. Factor profiles and their contributions to each factor identified by the positive matrix factorization model in BJ and GC. $b_{abs, total}$ and $b_{abs, BCpure}$ represent light absorption ($Mm^{-1}$) of coated BC particles and pure BC, respectively. BC types e.g., $BB_{pure}$ in units of count.

**Response to Reviewer #2**

The manuscript by Sun et al. investigated the chemical composition and mixing state of BC-containing particles and explored the driving factors of their evolution at an urban and a rural site. The results showed that $E_{abs}$ was highly dependent on the secondary inorganic aerosol coated on BC at both sites, while high primary coated OC also resulted in light absorption enhancement for relatively fresh BC particles at the rural site. $E_{abs}$ showed a similar evolutionary process at both sites: $E_{abs}$ was negligible in BC particles from fresh primary emissions and then increased significantly when BC is aged rapidly with increased coatings of OC-nitrate or nitrate. Finally, $E_{abs}$ could reach the highest value as sulfate involved in BC aging. This study novelly combined the mass concentration of chemical components with the peak area of individual particles to investigate the effect of the actual atmospheric chemical components on the black carbon mixing state as well as the light absorption enhancement. It is informative for exploring the relationship between the aerosol bulk composition and individual particle mixing state. I recommend it for publication on Atmospheric Chemistry and Physics after the authors consider several minor revisions to the manuscript.

Comments:

1. To better understand this study, it can be highlighted in section 2.1 that the SPAMS was deployed independently of TD. In section 3.3, section 3.4, the calculated optical properties of BC for Gucheng rural site could be comparable with a previous work conducted on a rural mountain site in Beijing (doi: 10.5194/acp-21-681-2021) while another study simultaneously performed both in IAP and on a mountain site (doi:10.1029/2020JD033096) may also be a good reference to your results on BC $E_{abs}$ considering one identical experimental location in both. Additionally, SOA playing an important role in BC absorption enhancement (Line 225) has been previously reported by an aircraft observation whereby more light cloud be reflected by clouds underneath BC aerosol layers resulting in more SOA formation, thus thicker coatings on BC (doi: 1088/1748-9326/ab4872). I suggest making more discussions on these according to the above works.

We thank the reviewers for providing additional studies to support our research which have been added and compared with our study in the new version of the manuscript. Following the reviewer's comments, we also added an instrument deployment on SPAMS in the method section.

"Previous studies found that SOA played an important role in BC absorption enhancement (Liu et al., 2019b), whereas in our study, the change in $E_{abs}$ seemed to be independent on (OC + metal)/$C_n$, likely because (OC + metal)/$C_n$ was influenced by both primary and secondary factors."

"One explanation is due to the variety of mixing state of BC from different sources, urban and rural background sites, and aging processes (Liu et al., 2015; Liu et al., 2017;

Liu et al., 2020)."

"As a result, $E_{abs}$ also presented a relatively consistent variation with FFOA and showed higher values at nighttime, indicating that the different chemical composition and mixing state associated with the changes in air masses due to the mountain valley winds had affected the light absorption enhancement of BC (Ding et al., 2021)."

"A high-resolution single particle aerosol mass spectrometer (HR-SPAMS, Hexin Instrument Co., Ltd.) was deployed independently in the same room for measuring the mixing state and chemical composition of single particles in ambient atmosphere."

2. line 79, "Ambient aerosols with a flow rate…, then aerosol particles ($\sim$2 L min$^{-1}$) were…" should be better divided into two sentences.

Changed as suggested.

3. line 103-104, "2 269 659 and 3 399 565 BC-containing particles are identified in Beijing and Gucheng, respectively." Should be "2 269 659 and 3 399 565 BC-containing particles were identified in Beijing and Gucheng, respectively."

Changed as suggested.

4. line 122, there is no need to emphasize Table S1.

We have deleted the description about Table S1 in this sentence.

5. Line 136, "suggesting that BC particles were more aged at the urban site." should be "suggesting that BC particles are more aged at the urban site."

Changed as suggested.

6. line 137-138, "which is twice than" should be "which was twice than", notice the consistent tense in a same sentence. Please check the full text.

Changed as suggested and checked the full text.

7. line 155, "… diesel vehicle emissions at the rural site especially nighttime." should be "… diesel vehicle emissions at the rural site especially at nighttime."

Changed as suggested.

8. line 173-174, "During the initial stage of haze episode (P0)" is better written as "During the initial stage of haze episode case1 (P0)".

Changed as suggested.

9. line 179, "Similar to haze episode 1" should be "Similar to case 1" which was mentioned in this study.

Changed as suggested.

10. line 198, ""Fig. S2" should be "Fig. S1"."

We have corrected the number of Figure.

11. line 212-213, "… quantify the impacts of POA and SOA on BC-coated OC." is better written as "… quantify the impacts of POA and SOA on OC coated on BC."

Changed as suggested.

12. line 242-243, "FactorB is the major type of aged BC in the rural area which was dominated by $BCOC_N$." should be "FactorB is the major type of aged BC in the rural area which is dominated by $BCOC_N$."

Changed as suggested.

13. line 253-254, "BC could be …, and lead to …" should be "BC could be …, and led to …", please check the tense of the full text.

We have changed as suggested and checked the full text.

14. line 280-281, "to above 1.30 after aged and internally mixed with nitrate" should be "to above 1.30 after aging and internally mixing with nitrate".

Changed as suggested.

15. line 96&253, "~" format should be consistent, please check the full text.

We have checked the full text and made all the formatting consistent.

16. Please keep the abbreviation of the single particle aerosol mass spectrometer consistent in this study (e.g. HR-SPAMS in line 89, SPAMS in line 205)

We checked the abbreviations throughout the text to make them consistent.

---

## Author Response (AR2)

**Response to the editor's comments**

We are thankful to the editor for the comments. We have revised the manuscript accordingly. Listed below is our point-to-point response in blue to each comment that was offered by the editor.

Comments to the author:

The authors have properly addressed the comments of the anonymous referees and they have modified their manuscript accordingly. However, alterations and corrections are needed for both the Main text and Supplement before the manuscript can be published in ACP:

Main text:

Line 13: Replace "Spectrometer" by "Spectrometry".

Lines 27, 60, and 285: Replace "in North" by "in the North".

Lines 28 and 288: Replace "showed much" by "showed a much".

Line 38: Replace "to soot and element" by "to as soot or elemental".

Line 39: Replace "of planetary" by "of the planetary".

Line 44: Replace "induced by" by "induced by the".

Line 48: Replace "while a" by "versus a".

Lines 49, 185, and 269: Replace "of mixing" by "of the mixing".

Line 51: Replace "used the" by "used".

Line 53: Replace "for characterization of size" by "for the characterization of the size".

Line 54: Replace "environment" by "environments" and replace "of soot" by "of the soot".

Line 58: Replace "more" by "a more".

Line 60: Replace "spectrometer (SPA-MS)" by "spectrometry (SPAMS)".

Line 61: Replace "most of BC" by "most BC".

Line 62: Replace "in heating" by "in the heating".

Line 64: Replace "winter particularly" by "winter, particularly".

Line 65: Replace "in NCP are" by "in the NCP, are".

Line 67: Replace "in NCP" by "in the NCP".

Lines 70, 202, 277, and 294: Replace "of BC" by "of the BC".

Line 71: Replace "in mixing" by "in the mixing".

Line 79: Replace "then, aerosol" by "then aerosol" and replace "the thermodenuder" by "a thermodenuder".

Line 80: Replace "and bypass" by "and a bypass".

Line 84: Replace "250 °C" by "250°C".

Line 87: Replace "in TD" by "in the TD".

Line 92: Replace "in ambient" by "in the ambient".

Line 96: Replace "inform" by "obtain".

Lines 96, 97, and 122: Replace "SPA-MS" by "SPAMS".

Line 105: Replace "by Computational" by "by the Computational" and replace "in MATLAB" by "in the MATLAB".

Line 108: Replace "with the peak areas" by "with peak areas".

Line 117: Replace "in positive" by "in the positive".

Line 121: Replace "are shown" by "are given".

Line 124: Abbreviations and acronyms should be defined (written full-out) when first used; therefore replace "bulk OA" by "bulk organic aerosol (OA)".

Line 128: Replace "PMF analysis" by "PMF analyses".

Line 130: Replace "of error matrix (in supplementary)" by "of the error matrix (in the Supplementary)".

Line 135: Replace "than that in GC (73%) yet higher than that" by "than in GC (73%) yet higher than".

Line 137: Replace "of aerosol" by "of the aerosol".

Line 138: Replace "spectra of" by "spectra of the".

Line 140: Replace "indicating that" by "indicating that the".

Line 141: Replace "80% of" by "80% of the".

Line 142: Replace "of secondary" by "of the secondary".

Line 143: Replace "that BC" by "that the BC".

Line 145: Replace "40% of" by "40% of the".

Line 167: Replace "that high" by "that a high".

Line 168: Replace "showing decreased" by "showing a decreased".

Lines 171, 295, and 298: Replace "as the increase" by "with the increase".

Line 174: Replace "in urban" by "in the urban".

Line 175: Replace "half of" by "half of the".

Line 176: Replace "in rural" by "in the rural".

Line 182: Replace "affect mixing" by "affect the mixing" and replace "with previous" by "with a previous".

Line 187: Replace "half day" by "half a day".

Line 188: Replace "during P1" by "during the P1".

Line 189: Replace "of haze" by "of the haze".

Line 191: Replace "causing high" by "causing a high".

Line 200: Replace "Similar variation" by "A similar variation".

Line 202: Replace "were similar" by "was similar".

We made all corrections as the editor suggested.

Line 203: The sentence part "which was characterized by most significant increase in $BCOC_N$ rather $BC_N$ as the PM increased" is unclear to me; it should be rephrased.

Following the editor's comment, we rewrote this sentence as: "Different from Beijing, the evolution of BC mixing state was similar during most of the haze events in GC (Figs. 2c and S3), which was generally characterized by a more significant increase in BCOCN than BCN."

Line 213: Replace "of total" by "of the total".

Line 215: Replace "between chemical" by "between the chemical".

Line 216: Replace "we summed" by "we summed the".

Line 217: Replace "represent secondary" by "represent the secondary" and replace "of positive" by "of the positive".

Line 219: Replace "in total" by "in the total".

Line 220: Replace "between peak" by "between the peak" and replace "and mass" by "and the mass".

Line 221: Replace "As the increase" by "With the increase".

Line 223: Replace "of haze" by "of the haze".

Line 226: Abbreviations and acronyms should be defined (written full-out) when first used; therefore replace "POA" by "primary OA (POA)".

Line 228: Replace "than SIA at both" by "than that of secondary inorganic aerosol at both the".

Line 233: Replace "Light" by "The light".

Line 235: Replace "in BC" by "in the BC".

Line 238: Replace "were likely" by "likely".

Line 239: Replace "As the progress" by "With the progress".

Line 242: Replace "was closed to 0. Pervious" by "was close to 0. A previous".

Line 245: Replace "in atmosphere" by "in the atmosphere".

We made all corrections as the editor suggested.

Line 251: It is unclear what "CAPS" in "TD-CAPS" is; it should be written full-out.

Thank the editor's comment. It was spelled out in the revised manuscript: "We also estimated the $E_{abs}$ in summer 2017 using the same method and compared with the measurements by a cavity attenuated phase shift single scattering albedo monitor coupled with a thermodenuder (Fig. S9)."

Line 252: Replace "average about" by "average of about".

Lines 255 and 256: Replace "mixing state" by "mixing states".

Line 256: Replace "respectively to" by "respectively, to".

Line 261: Replace "and internally" by "and became internally".

Line 272: Replace "led to an" by "lead to an".

Line 276: Replace "The detail" by "The detailed" and replace "in supplementary" by "in the Supplementary".

Line 279: Replace "of radiative" by "of the radiative".

Line 282: Replace "on radiative" by "on the radiative".

Line 289: Replace "with large" by "with a large" and replace "of clean" by "of the clean".

Line 295: Replace "at both" by "at both the".

Line 298: Replace "in the increase of" by "in the increase of the".

Line 299: Replace "particles in" by "particle type in".

Line 300: Replace "different mixing" by "a different mixing".

Line 301: Replace "internally mixed" by "internal mixing".

Line 371: Replace "amp, apos, Connor" by "O'Connor".

Line 399: Replace "Allen, James D." by "Allen, J. D.".

Line 505: Replace "The digital" by "Digital".

Line 507: Replace "in mass" by "in the mass".

Line 508: Replace "in individual" by "in the individual".

Line 522: Replace "percentiles of" by "percentiles of the".

We made all corrections as the editor suggested.

Supplement:

Page 1, line 13: Replace "Mass Spectrometer" by "Mass Spectrometry".

Page 2, line 1: Replace "PMF" by "Positive matrix factorization (PMF)".

Page 2, line 2: Replace "state on Eabs" by "states on light absorption enhancement (Eabs)" and replace "of PMF" by "of the PMF".

Page 2, line 3: Replace "particles," by "particles (for the definitions of babs, total, babs, BCpure and BC, see the main text)

Page 2, line 2 below equation (1): Replace "in j row. LODs" by "in the jth row. The LODs".

Page 2, line 4 below equation (1): Replace "PMF solutions" by "the PMF solutions".

Page 2, line 5 below equation (1): Replace "mixing state" by "mixing states".

Page 2, line 9 below equation (1): Replace "by pervious" by "by a previous".

Page 2, line 4 below equation (2): Replace "Wavelength" by "The wavelength".

Page 2, line 6 below equation (2): Replace "is effective" by "is the effective".

Page 2, line 7 below equation (2): Replace "between aerosol" by "between the aerosol".

Page 2, line 8 below equation (2): Replace "and light" by "and the light" and replace "from PAX" by "from a photoacoustic extinctiometer (PAX))".

Page 3, heading of Table S1: Replace "A summary of abberviations and descrptions" by "Summary of abbreviations and descriptions".

Page 3, in the right column of Table S1: replace "Dall et al., 2012" by "Dall'Osto and Harrison, 2012".

Page 3, heading of Table S2: Replace "A summary of relationship between aerosol optical depth and light" by "Summary of the relationship between the aerosol optical depth and the light"

We made all corrections as the editor suggested.

Page 3, Table S2: The first column is unclear. I presume that the unit of the "Effective Height" is in m; but what does "slope" do after m? And what does "r" denote? This first column should be modified.

Following the editor's suggestion, we revised Table S2 (replaced by Table R1), and added the description of the first column.

Table R1. Summary of the relationship between the aerosol optical depth and the light extinction coefficient measured by PAX at both sites. The slope represents the effective height and r represents the correlation coefficient.

|  | Beijing | Gucheng |
| --- | --- | --- |
| Effective Height (m) | 711 | 554 |
| $r$ | 0.73 | 0.51 |

Page 4, Figure S1: It is unclear to me what "ramian" in the ordinate denotes.

It should be "remaining". We corrected the typo in the revised manuscript.

[Figure]

Figure R1. Loss of pure BC particles in the thermodenuder under different temperatures.

Page 4, caption of Figure S1: Replace "The loss of pure BC particles in TD" by "Loss of pure BC particles in the thermodenuder".
Page 5, last line: Replace "in BJ (left panel) and GC" by "in Beijing (left panel) and Gucheng (GC)".
Page 6, caption of Figure S3: Replace "in BJ and GC" by "in Beijing (BJ) and Gucheng (GC)".
Page 6, first line of the caption of Figure S4: Replace "The relationship between the sum of mass" by "Relationship between the sum of the mass".
Page 6, second line first line of the caption of Figure S4: Replace "in GC" by "in GC (for the definitions of NR-PM1 and eBC, see the main text)".
Page 7, caption of Figure S5: Replace "PM, FFOA and BBOA in Case2" by "NR-PM1, FFOA and BBOA in Case2 (with FFOA standing for fossil fuel-related organic aerosol (OA) and BBOA for biomass burning OA)".
We made all corrections as the editor suggested.

Page 7, caption of Figure S6: Replace "Eabs" by "light absorption enhancement (Eabs)".
Page 8, caption of Figure S7: Replace "in GC)." by "in GC); for the definitions of OC, SOA, eBC LO-OOA, MO-OOA, OOA and aq-OOA, see the main text.".
They were corrected following the editor's comments.

Page 9, caption of Figure S9: Replace "The comparisons" by "Comparison".
Page 10, line 1: replace "Dall, amp, apos, Osto, M., and Harrison" by "Dall'Osto, M. and Harrison".
Corrected as suggested.

---

## Author Response (AR3)

**Response to the editor's comments**
We are thankful to the editor for the comments. We have revised the manuscript accordingly. Listed below is our point-to-point response in blue to each comment that was offered by the editor.

A few alterations and corrections are still needed for both the Main text and Supplement before the manuscript can be published in ACP:

Main text:
Line 129: Replace "detailed PMF analysis" by "detailed PMF analyses".
Line 130: Replace "analyses was also" by "analysis was also".
Lines 203 and 280: Replace "of BC" by "of the BC".
We made all corrections as the editor suggested.

Supplement:
Page 2, line 2: Replace "mixing state on" by "mixing states on".
Page 2, line 4: Replace "thermodenuder time (TD)" by " thermodenuder (TD) time".
Also is the word "time" needed here or should it perhaps be replaced by "line"?
Page 2, line 5: Replace "that was defined" by "was defined".
Page 8, caption of Figure S7, last line: Delete the " before the terminal period (.).
We made all corrections as the editor suggested.